# CO₂- and orbitally- driven oxygen isotope variability in the Early Eocene

Julia Campbell[1], Christopher J. Poulsen[1,2], Jiang Zhu[3], Jessica E. Tierney[4], and Jeremy Keeler[1]

[1]Department of Earth and Environmental Sciences, University of Michigan, Ann Arbor, 48104, USA
[2]Department of Earth Sciences, University of Oregon, Eugene, 97403, USA
[3]Climate and Global Dynamics Laboratory, National Center for Atmospheric Research, Boulder, 80301, USA
[4]Department of Geosciences, University of Arizona, Tucson, 85721, USA

*Correspondence to*: Julia Campbell (juliacam@umich.edu)

**Abstract.** Paleoclimate reconstructions of the Early Eocene provide important data constraints on the climate and hydrologic cycle under extreme warm conditions. Available terrestrial water isotope records have been primarily interpreted to signal an enhanced hydrologic cycle in the Early Eocene associated with large-scale warming induced by high atmospheric $CO_2$. However, orbital-scale variations in these isotope records have been difficult to quantify and largely overlooked, even though orbitally driven changes in solar irradiance can impact temperature and the hydrologic cycle. In this study, we fill this gap using water isotope-climate simulations to investigate the orbital sensitivity of Earth's hydrologic cycle under different $CO_2$ background states. We analyze the relative difference between climatic changes resulting from $CO_2$ and orbital changes and find that the seasonal climate responses to orbital changes are larger than $CO_2$-driven changes in several regions. Using terrestrial $\delta^{18}O$ and $\delta^2H$ records from the Paleocene-Eocene Thermal Maximum (PETM), we compare our modeled isotopic seasonal range to fossil evidence and find approximate agreement between empirical and simulated isotopic compositions. The limitations surrounding the equilibrated snapshot simulations of this transient event and empirical data include timing and time-interval discrepancies between model and data, the preservation state of the proxy, analytical uncertainty, the relationship between $\delta^{18}O$ or $\delta^2H$ and environmental context, and vegetation uncertainties within the simulations. In spite of the limitations, this study illustrates the utility of fully coupled, isotope-enabled climate models when comparing climatic changes and interpreting proxy records in times of extreme warmth.

## 1 Introduction

The Earth has rapidly warmed since the preindustrial (PI) era, driving substantial and widespread changes in the hydrologic cycle (Douville et al., 2021). Severe warming and changes in the water cycle are projected to continue depending on the level of greenhouse gas emissions. Following a higher emissions pathway, atmospheric $CO_2$ will exceed 1,000 ppm by the end of the 21st century, a level that last existed during the Early Eocene about 56-48 million years ago (Tierney et al., 2020). By modeling different orbital and $CO_2$ configurations of the Early Eocene, and matching the simulations to fossil evidence, we can provide context for the proxy records and learn how the orbit may have played a part in the severe warming at the onset

of the Paleocene-Eocene Thermal Maximum (PETM), as well as different seasonal impacts on the Early Eocene climate. This study serves to distinguish the warming signal from the orbital signal within the hydrologic cycle under the most recent extreme

warmth.

The Early Eocene likely experienced atmospheric $CO_2$ concentrations at 3x PI levels, though proxies range from 500 to 2000 ppm (Anagnostou et al., 2020; Rae et al., 2021). The PETM was a ~100,000-year time interval within the Early Eocene that experienced especially rapid warming with global surface temperatures rising 5-6°C in response to an increase in $CO_2$ levels

as high as 6x PI $CO_2$ (Zhu et al., 2019; Inglis et al., 2020; Tierney et al., 2022). Although the geography of the Eocene differs from modern-day geography, simulation of this warming event furthers our understanding of how the Earth operates under a high $CO_2$ background state. There is still much to learn about the Early Eocene and the PETM, especially surrounding the relative influence changes in orbit have on the hydrologic cycle.

Earth's orbital configuration has a strong influence on regional climate and is a driver of major climatic fluctuations (Davis and Brewer, 2011). The orbit determines the timing and intensity of sunlight for a given region and season. Obliquity, the tilt on Earth's axis, has a ~41,000-year cycle; precession, the Earth's wobble, is a ~22,000-year cycle; and eccentricity, the Earth's path around the sun, lasts ~100,000 years. These three factors together determine the solar irradiance any area on Earth will receive at a given time. A higher eccentricity and a higher obliquity cause heightened seasonality, including warmer summer

seasons (Davis and Brewer, 2011). Warmer summers often melt more ice which can accelerate a climatic fluctuation, but the warm Early Eocene lacks a cryosphere, which may have modified the climate's response to warmer summers. Seasonal shifts in solar insolation also drive temperature changes which impact the isotopes of precipitation, and therefore the proxy records related to the isotopic composition of precipitation.

There is evidence of orbital-scale variations in atmospheric $CO_2$ during the Late Paleocene and Early Eocene (Zeebe et al., 2017). Orbitally induced changes in the oceanic temperatures and circulation may have also been a cause for the frequent and variable hyperthermals during the Early Eocene (Lunt et al., 2011; Piedrahita et al., 2022). The PETM was the most extreme hyperthermal, a consequence of even greater atmospheric $CO_2$ concentrations and potentially greater seasonal changes owing to the Earth's orbit.


Oxygen and hydrogen isotopic ratios from meteoric waters are often used as a measure of climate variability, including variability by changing $CO_2$ concentrations or orbit. The ratio of heavy ($^{18}O$, $^2H$) to light ($^{16}O$, $^1H$) isotopes is represented by $\delta^{18}O$ or $\delta^2H$ (Craig, 1961). Warmer global temperatures mean more energy in the troposphere to increase evaporation of the heavier isotope – commonly referred to as the temperature effect, and decreased precipitation results in rainfall more enriched

in the heavier isotope – commonly referred to as the amount effect, which increases the atmospheric $\delta^{18}O$ or $\delta^2H$ (Craig, 1961). This can be linked to orbit and atmospheric $CO_2$ because distribution of solar insolation and greenhouse gas concentrations

each drive global temperature and precipitation trends, which then influence the $\delta^{18}O$ or $\delta^2H$ of precipitation ($\delta^{18}O_p$, $\delta^2H_p$) (Craig, 1961). At a regional scale, oceanic and atmospheric dynamics, like flow of air mass transport and origin of evaporated water, also impact water isotopes.


Although the Early Eocene and PETM have been modeled before, these are the first simulations of that time period to reproduce the extreme warmth and weakened meridional temperature gradient of the PETM, as well as track water isotopes through the hydrologic cycle – which can offer more information on evaporation, advection, precipitation, and other factors that influence $\delta^{18}O_p$ and $\delta^2H_p$ (Zhu et al., 2020). These simulations track water isotopes in every component of the model (Brady et al., 2019).

This Early Eocene model has been previously published in Zhu et al (2019), Zhu et al (2020), and Tierney et al (2022). The simulations at varied $CO_2$ have been used to analyze ocean circulation and shortwave cloud feedbacks to further understand parameterizations within the model that play a role in large-scale climate sensitivity (Zhu et al., 2019; Zhu et al., 2020). Additionally, these simulations have been statistically sampled through a data assimilation approach in order to reconstruct PETM climate changes (Tierney et al., 2022). None of these previous studies investigate the exceptional variations in seasonal

climate between orbits or the sensitivity of the terrestrial water cycle to both orbital and atmospheric $CO_2$ changes. Through tracking $\delta^{18}O_p$, this paper underlines the importance of orbital cycles in understanding terrestrial water isotopes.

The biosphere has an impact on water isotopes as well, especially since vegetation patterns likely shifted from the PETM to the Early Eocene. Precipitation infiltrates the soil, is taken up by the roots of plants, and then transpired through leaves.

Different plants exhibit different preferential fractionation of water isotopes, which leads to different isotopic signals in the transpired moisture returning to the atmosphere (Gat and Airey, 2006). Plant water oxygen isotope signals can shift by about 6.5‰ between the rainy and dry seasons, primarily due to shifts in soil water isotopes (Dai et al., 2020). Quantitative constraints on the plant isotopic effect on atmospheric moisture and precipitation remain difficult to obtain, even more so on a global or regional scale (Gat, 2000). In addition, the vegetation changes for the PETM to Early Eocene are not well constrained, and the

estimated fractionation factors for Early Eocene vegetation have high uncertainty (Sachse et al., 2012). Finally, the isotope-enabled land model equipped here assumes the transpired water has the same isotope ratio as the root-weighted soil water (Brady et al., 2019). Therefore, the way water isotopes interact in the biosphere-atmosphere space is primarily based off the soil water, rather than various plant types. The isotope ratio of leaf water is set by the requirement of isotopic mass balance within the plant. There would not be a significant impact on atmospheric water isotope ratios with different vegetation types

within the model, as long as the vegetated areas remained vegetated. To that, these simulations use identical vegetative inputs for each run, isolating $CO_2$ and orbit as controlling factors on changes in water isotopes.

In this study, we use sensitivity experiments to investigate the Early Eocene's climatic and hydrologic response to changes in Earth's orbit and atmospheric $CO_2$ concentration to further understand the impact of orbit on the hydrologic cycle during warm

climates and to test Earth's sensitivity to changes in orbit under different $CO_2$ background states. Additionally, we include

comparisons between these responses and terrestrial $\delta^{18}O$ and $\delta^2H$ records to test the model's ability to simulate changes in the hydrologic cycle in an extremely warm climate. The terrestrial data is not constrained to a specific orbit or time of year, so we compare the data to all orbits simulated and the entire seasonal range to determine if the global water isotope signal is captured by the model and to tease out any potential seasonal biases in the data. These analyses strengthen our comprehension

of the environmental context of terrestrial proxy records, the potential of orbital changes to partly initiate the hyperthermal, the influence of orbit on the hydrologic cycle at different $CO_2$ forcings, and the potential of this model to simulate climate changes during a time with a higher atmospheric $CO_2$ level than today. We largely find that the orbit may have a more substantial impact on terrestrial water isotope records than atmospheric $CO_2$ in certain regions, particularly in seasonally biased datasets.


## 2 Methodology

### 2.1 Earth system modeling

The simulations were conducted using a water isotope-enabled Community Earth System Model (iCESM) version 1.2. CESM1.2 is comprised of the Community Atmosphere Model (CAM) version 5.3, Community Land Model (CLM) version

4.0, Community Ice Code (CICE) version 4.0, River Transport Model (RTM), Parallel Ocean Program (POP) version 2, and a coupler to connect them (Hurrell et al., 2013). There are 30 vertical levels in the atmosphere and 60 vertical levels in the ocean. In addition, iCESM has the capability to simulate the transport and transformation of water isotopes in the model hydrologic cycle (Brady et al., 2019). Although there are some noticeable biases in iCESM's isotope tracking, such as a slight depleted bias in $\delta^{18}O_p$, the model captures the general quantitative features of water isotope movement (Brady et al., 2019).

The model resolution is rather coarse at 1.9° x 2.5° for the atmosphere and land, and a nominal 1° for ocean and sea ice. For this reason, we focus on analyzing large-scale, global patterns in the hydrological cycle. The paleogeography, land-sea mask, and vegetation distribution follow the Deep-Time Modeling Intercomparison Project (DeepMIP) protocol at about 55 million years ago (Herold et al., 2014). The ocean temperature and salinity were initialized from a PETM quasi-equilibrated state, and the $\delta^{18}O$ of seawater was initialized from a constant -1‰ to account for the absence of ice sheets in a hothouse climate (Zhu

et al., 2020). We completed eight experiments, a control simulation with a modern orbit (OrbMod) and 3x PI $CO_2$, and seven sensitivity experiments with differing orbital configurations and $CO_2$ levels. There are four orbital configurations, each run at both a low (3x PI) and high (6x PI) $CO_2$ concentration, including a modern orbit (OrbMod), maximum summer solar insolation for the Southern Hemisphere (OrbMaxS), maximum summer solar insolation for the Northern Hemisphere (OrbMaxN), and minimum eccentricity (OrbMin) (Lunt et al., 2011). The OrbMaxS and OrbMaxN simulations experience high eccentricity

and obliquity, so those simulations would expect high seasonality (Lunt et al., 2011). A control simulation was run for 2000 model years. Orbital cases were branched from the control simulation and run for an additional 500 model years each. The climatological means presented in the results are based on the last 100 years of each simulation. Mean annual climatologies

can be found in Figs. A11-A13. All simulations had an atmospheric $CH_4$ concentration of 791.6 ppb and an atmospheric $N_2O$ concentration of 275.68 ppb. Atmospheric greenhouse gases other than $CO_2$, like $CH_4$, may have been higher in a warmer

climate, but these are kept identical between simulations. Greenhouse gases other than $CO_2$ are poorly constrained for the Early Eocene and the warming effect on the water cycle is largely captured by the change in atmospheric $CO_2$. $CO_2$ and orbital details for the modeled cases discussed here are available in Table 1. Further details are available in Tierney et al (2022).

| | 3x PI OrbMod | 3x PI OrbMin | 3x PI OrbMaxN | 3x PI OrbMaxS | 6x PI OrbMod | 6x PI OrbMin | 6x PI OrbMaxN | 6x PI OrbMaxS |
|---|---|---|---|---|---|---|---|---|
| $CO_2$ (ppm) | 854.1 | 854.1 | 854.1 | 854.1 | 1708.2 | 1708.2 | 1708.2 | 1708.2 |
| Eccentricity | 0.0167 | 0.0 | 0.054 | 0.054 | 0.0167 | 0.0 | 0.054 | 0.054 |
| Obliquity | 23.45 | 22 | 24.5 | 24.5 | 23.45 | 22 | 24.5 | 24.5 |
| Moving Vernal Equinox Longitude of Perihelion | 90 | 0 | 270 | 90 | 90 | 0 | 270 | 90 |

**Table 1: The atmospheric $CO_2$ concentrations and orbital details are presented here for each modeled case. OrbMod represents a modern orbit, OrbMin represents a minimum eccentricity orbit with mild seasons, and OrbMaxN and OrbMaxS represent maximum eccentricity orbits with heightened seasonality. The longitude of the perihelion is the angle between the Earth at the Northern Hemisphere (NH) autumnal equinox and the Earth at its closest to the sun – $0°$ represents a perihelion at the NH autumnal equinox, $90°$ is at the NH winter solstice, $180°$ is at the NH vernal equinox, and $270°$ is at the NH summer solstice. For example, the NH is closest to the sun during the NH summer solstice in OrbMaxN (Fig. A1).**


**2.2 Proxy records**

There are more hydrological isotope proxy data from the PETM, so that time interval is the focus here, rather than the Early Eocene. The 6x PI $CO_2$ simulations are compared to several terrestrial proxy records of $\delta^{18}O$ and $\delta^2H$ from the PETM (Table

2). Our focus is on terrestrial proxies over marine proxies as terrestrial proxies are more heavily impacted by changing orbit and seasonality, given the land-ocean warming contrast (Byrne and O'Gorman, 2013). However, the terrestrial data is not dated to a specific orbit or season given uncertainties in the dating relative to orbital pacing, so we use it as an approximate envelope of PETM water isotope values against the range of values from all simulated orbits and seasons. Zhu et al previously showed good model-data agreement between terrestrial temperature proxies and the Early Eocene (3x PI $CO_2$) simulation,

validating the model's performance (Fig. A2, Zhu et al., 2019). These Eocene simulations also show strong agreement with marine proxies, as discussed in Zhu et al (2020). The suite of Eocene simulations run as part of the Paleoclimate Modeling Intercomparison Project exhibit close agreement with terrestrial precipitation proxies as well (Cramwinckel et al., 2023).

Paleosol carbonates and siderite preserve the isotopic signal of the soil water they form in, thus a comparison with simulated soil water, rather than precipitation, is the most salient comparison. Soil carbonates are likely to form around 40-100 cm deep, so the simulated $\delta^{18}O$ range represents soil water at this depth as well (Burgener et al., 2016). In order to compare the proxy $\delta^{18}O$ to simulated soil water $\delta^{18}O$, a proxy system model (PSM) is needed to transform $\delta^{18}O$ of the soil carbonate to $\delta^{18}O$ of the water in which they precipitated from. The fractionation factor is controlled by the local soil temperature, so each location has a slightly different fractionation factor, though all are near 1 (van Dijk et al., 2018; Friedman and O'Neil, 1977). Siderite oxygen fractionation was calculated through Van Dijk's best fit equation (van Dijk et al., 2018). The paleosol carbonate's oxygen fractionation was calculated through the USGS equation (Friedman and O'Neil, 1977).

Leaf waxes are another powerful tool for paleoclimate reconstruction and also require a PSM to account for the transformations between the $\delta^2H$ of soil water and the $\delta^2H$ within the leaf wax (Konecky et al., 2019). Therefore, the $\delta^2H$ model-data comparison includes $\delta^2H$ from PETM leaf waxes and model-inferred leaf wax $\delta^2H$ seasonal ranges. Comparing leaf wax $\delta^2H$ in addition to the soil $\delta^{18}O$ proxies offers an opportunity to explore hydrogen isotope accuracy within iCESM. The model-inferred leaf wax $\delta^2H$ values were calculated through the WaxPSM using the zonal seasonal range of soil water $\delta^2H$ and a global, fixed apparent fractionation factor of -124‰ (Konecky et al., 2019). The apparent fractionation factor differs between different plants, but it is unknown for Early Eocene vegetation, so this study uses an average value for a modern landscape that is equal parts shrubs, trees, forbs, and C3 grasses (Sachse et al., 2012).

| Author (Year) | Sample Type | Age |
|---|---|---|
| Bataille et al (2016) | Paleosol Carbonate, $\delta^{18}O$ | PETM |
| Bowen et al (2015) | Paleosol Carbonate, $\delta^{18}O$ | PETM |
| Kelson et al (2018) | Paleosol Carbonate, $\delta^{18}O$ | PETM |
| Koch et al (1995) | Paleosol Carbonate, $\delta^{18}O$ | PETM |
| Snell et al (2013) | Paleosol Carbonate, $\delta^{18}O$ | PETM |
| White et al (2017) | Siderite, $\delta^{18}O$ | PETM |
| Van Dijk et al (2020) | Siderite, $\delta^{18}O$ | PETM |
| Pagani et al (2006) | Leaf Wax, $\delta^2H$ | PETM |
| Handley et al (2008) | Leaf Wax, $\delta^2H$ | PETM |
| Handley et al (2011) | Leaf Wax, $\delta^2H$ | PETM |
| Smith et al (2007) | Leaf Wax, $\delta^2H$ | PETM |

| Tipple et al (2011) | Leaf Wax, $\delta^2$H | PETM |
| --- | --- | --- |
| Garel et al (2013) | Leaf Wax, $\delta^2$H | PETM |
| Jaramillo et al (2010) | Leaf Wax, $\delta^2$H | PETM |
| Huber and Caballero (2011) | Macroflora, Temperature | Early Eocene |

**Table 2: A list of the proxy records used in model-data comparisons. Paleosol carbonates and siderite are used for a PETM oxygen isotope comparison, and leaf fossils are used for a PETM hydrogen isotope comparison. A terrestrial temperature comparison, reconstructed from macroflora fossil evidence, is used to validate the Early Eocene (3x PI CO₂) simulations in Fig. A2.**


The soil carbonate proxies act as a time-integrated record of environmental changes over hundreds to thousands of years, while the leaves form over a matter of weeks (Burgener et al., 2016). Although the proxy records span a wide geographic range, we could not find published terrestrial $\delta^{18}$O proxy records for the PETM from the Southern Hemisphere (SH), which signals a need for more focus and funding to be directed to researchers in the SH to fill this gap. The locations of the proxies were

converted to paleo-coordinates suitable for our paleogeographic reconstruction (Fig. 1, Muller et al., 2018).

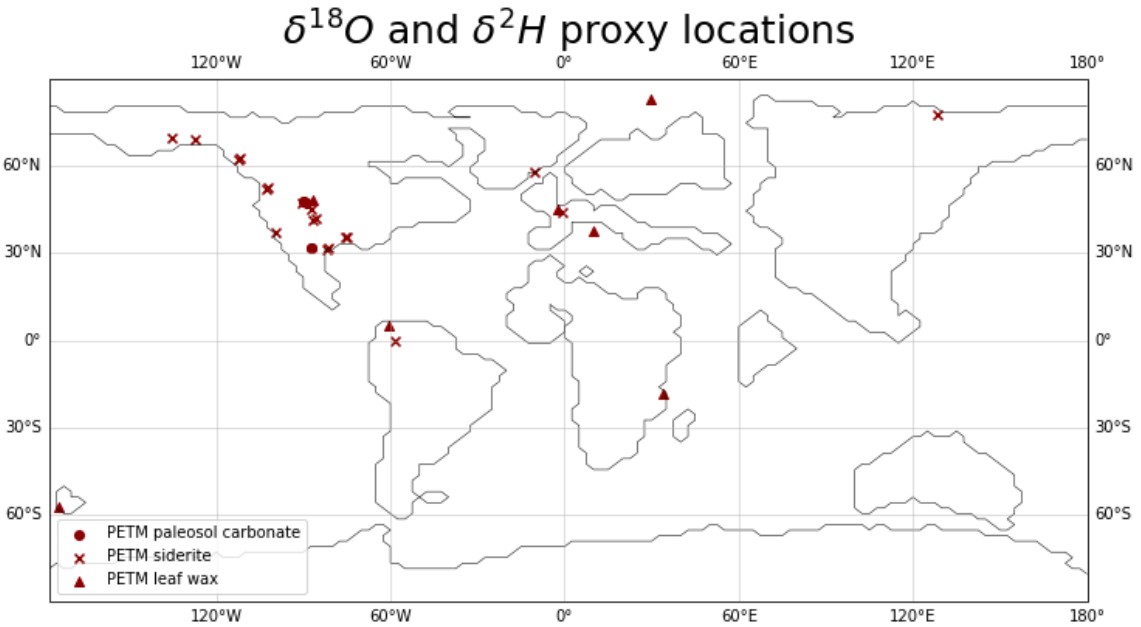

**Figure 1: The Early Eocene paleogeography used for the model simulations, with points overlaid representing the locations of the PETM terrestrial isotope proxy record. Note that two of the records originated on islands that don't appear on this paleogeographic map. There is a noticeable lack of paleo-isotope records from the SH.**


**3 Results**

### 3.1 Response to orbit


High eccentricity and obliquity cause more intense seasonality (Berger, 1988). We focus our analysis on the OrbMaxS and OrbMaxN simulations because their large seasonal insolation variations drive the greatest terrestrial climate response (Fig. A1). OrbMaxS and OrbMaxN are at the same (high) eccentricity and obliquity as one another, so these differences are the result of precessional changes only (Table 1). Sect. 3.1 focuses on response to orbit at the 3x PI $CO_2$ level because there were

greater climatic differences between orbits at this level (see Sect. 3.3).

Differential solar heating, controlled by orbital forcing, gives rise to a latitudinal temperature gradient and has a great influence on Earth's climate. Insolation changes drive differences in specific humidity between these runs, since warmer air can hold more water vapor with an increase in saturation vapor pressure (SVP) (Fig. A4). However, areas that experience an increase in temperature and specific humidity, like central Africa or Australia during DJF, largely experience a decrease in relative

humidity, the ratio of water vapor to the SVP, as SVP increases more than specific humidity during warming (Fig. A5, Tichy et al., 2017). Lower relative humidity results in a lower chance of cloud formation and rainfall, which often only occurs when relative humidity is >90%. With a lower chance of precipitation, these areas are less likely to experience isotopic rainout of the heavy isotope and therefore often exhibit higher $\delta^{18}O_p$ (Figs. 3, 4).


Both $\delta^{18}O_p$ and $\delta^{2}H_p$ exhibit large-scale seasonal differences between OrbMaxS and OrbMaxN; DJF (JJA) experiences greater (lesser) $\delta^{18}O_p$ and $\delta^{2}H_p$ in OrbMaxS than OrbMaxN driven by the difference in seasonal insolation (Figs. 4, A6). The increase in insolation during DJF also encourages stronger evaporation rates from the sea surface, which is influential on isotopic signals as the origin of transported moisture, and sometimes encourages continental recycling through evapotranspiration (Figs.

4, A16; Gierz et al., 2017; Risi et al., 2019). With higher insolation and lower relative humidity, the air is generally drier and able to stimulate higher rates of evaporation (Fig. A5). Therefore, the DJF season experiences higher insolation, warmer temperatures, higher rates of evaporation, and generally a stronger presence of heavier isotopes in atmospheric moisture and, in turn, precipitation (Figs. 2, 3, 4, A3, A5). JJA experiences a decrease in insolation, lower temperatures, lower rates of evaporation, and generally a weaker presence of heavy isotopes in precipitation (Figs. 2, 4, A1, A5). Globally, the large-scale

simulated $\delta^{2}H_p$ patterns mimic the large-scale simulated $\delta^{18}O_p$ patterns, since the fractionation of hydrogen and oxygen are controlled by the same distillation factors (Fig. A6).

Aside from the large-scale isotopic signals, specific regions experience large seasonal differences in $\delta^{18}O_p$ with a change in orbit. For instance, western North America sees a substantial [18]O-depletion in precipitation during JJA when comparing

OrbMaxS to OrbMaxN, up to -5‰ (Fig. 4). JJA is the SH winter season, so most of the Earth is colder at this time in the OrbMaxS simulations, especially regions of high elevation like the North American Cordillera, which likely had an elevation upwards of 3 km in some areas (Fig. 2, Mulch et al., 2007). The decrease in temperature resulted in an increase in relative

humidity in the area, increasing rainfall (Figs. 3, A5). Mountains already intensify vapor fractionation, reducing the $\delta^{18}O_p$ through the amount and temperature effects, so the Cordillera experienced especially depleted rainfall during JJA for the

OrbMaxS simulations compared to the OrbMaxN simulations, which experienced a very hot NH during JJA (Fig. 4, Poulsen et al., 2007). Evaporated water from the cool Pacific Ocean travels in the prevailing westerly winds over continental North America. As the moisture ascends the mountainside, there is increased rainfall, further depleting the clouds of the heavier water isotopes, and leading to a dry and isotopically light descending air mass (Fig. 4). Furthermore, regions like northern Africa or the Tibetan Plateau experience increases in temperature and decreases in precipitation year-round, as well as increases

in $\delta^{18}O_p$ mostly due to temperature and amount effects (Figs. 2, 3 4). Northern Africa experiences enriched rainfall during all seasons, especially SON, with differences up to ~6.5‰ between OrbMaxS and OrbMaxN (Fig. 4). Temperatures are warmer, and relative humidity and rainfall rates are lower, resulting in substantially enriched rainfall (Figs. 2, 3, 4, A5). High rates of evaporation from the warm pool accompany the trade winds to transport relatively isotopically heavy moisture to the primarily warm and dry Sahara Desert region (Figs. A5, A16). This region has sparse vegetation and resulting low rates of

evapotranspiration, so the water isotopes in precipitation are largely consequence of the evaporative source – the nearby seawater. The cooler, drier wind above the Indo-Pacific during SON passes over the warm sea and evokes higher evaporation rates due to the strong gradients in temperature and moisture between the air-sea surfaces, and that enriched air mass is quickly swept away towards the nearby land mass. The little rainfall this region experiences is therefore isotopically heavier in the OrbMaxS simulation (Fig. 4).


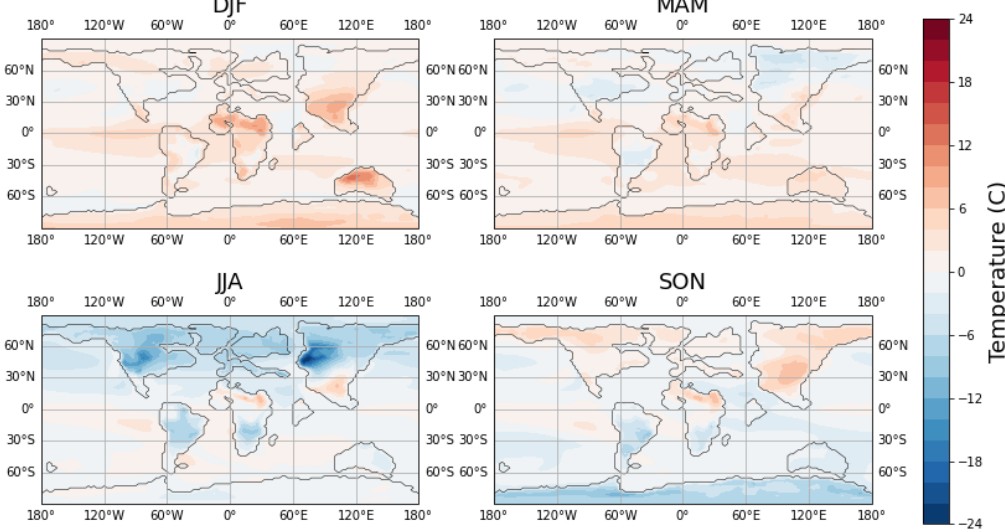

**Figure 2: The seasonal temperature differences between OrbMaxS and OrbMaxN at 3x PI CO$_2$.**

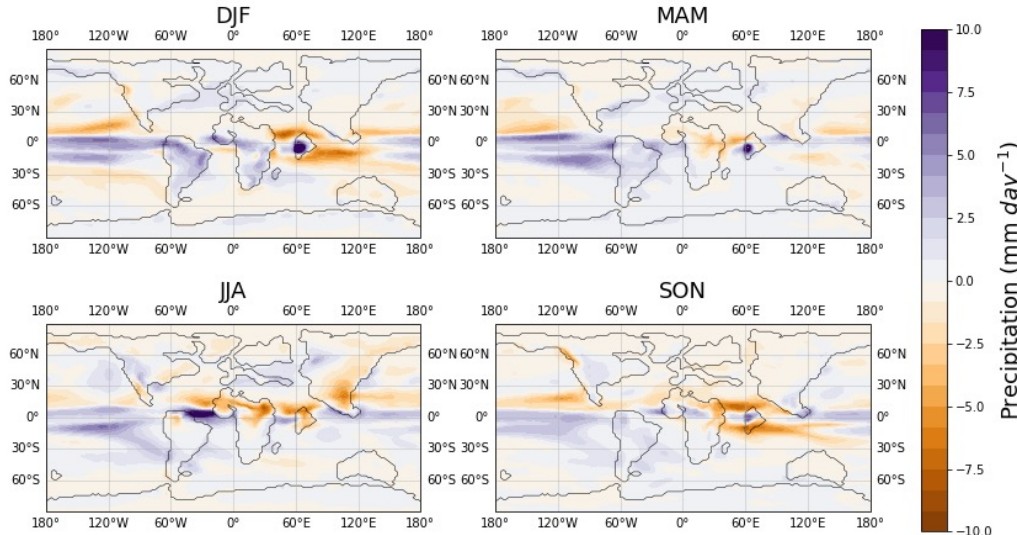

**Figure 3: The seasonal precipitation differences between OrbMaxS and OrbMaxN at 3x PI CO₂. The maximum positive precipitation difference on the corresponding color bar represents anything experiencing 10 mm day⁻¹ or higher.**

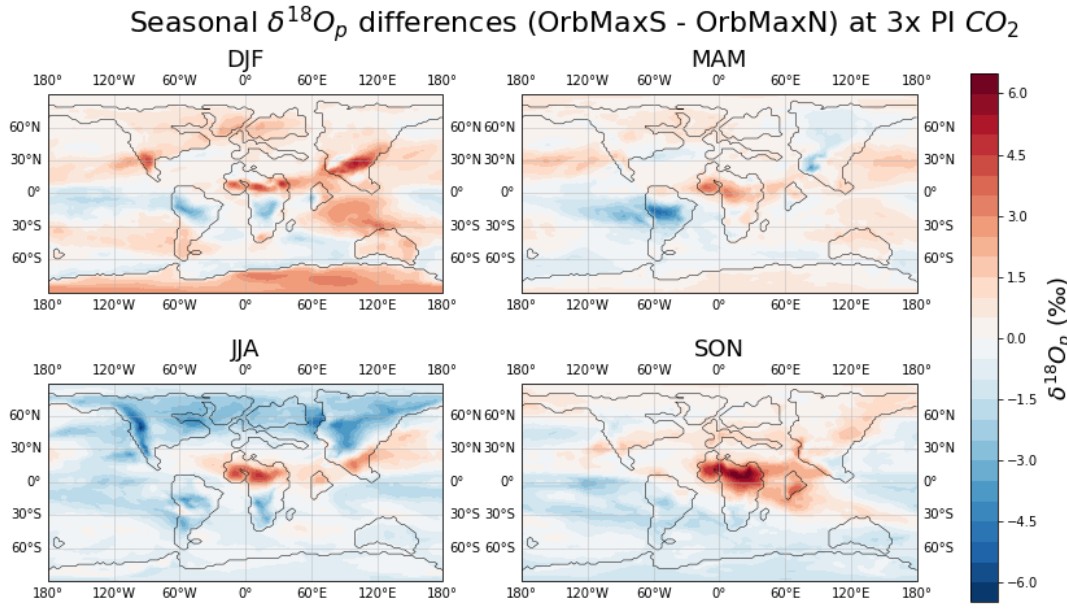

**Figure 4: The seasonal $\delta^{18}O_p$ differences between OrbMaxS and OrbMaxN at 3x PI CO₂.**

## 3.2 Response to CO₂

Past studies have shown that simulated climate responds strongly to changes in atmospheric $CO_2$ levels (Poulsen et al., 2007). Here, we compare the climatic $CO_2$ response for different orbits. Increasing greenhouse gases results in warmer global temperatures, an enhanced hydrologic cycle, and $^{18}O$ enrichment over land (Figs. 5, 7). Sect. 3.2 shows hydrological and $\delta^{18}O_p$ responses to increased $CO_2$ for both OrbMaxS and OrbMaxN. Seasonal partitioning did not reveal any further insights, so we focus on annual-average results.

As a consequence of the increase in $CO_2$, globally averaged surface temperatures increase by ~6°C in the 6x PI $CO_2$ simulations compared to the 3x PI $CO_2$ simulations (Fig. 5). There is a greater rise in temperature over land than ocean – an average 41-44% increase in surface temperature over land compared to an average 17-19% increase in surface temperature over ocean, largely owing to the land-sea contrast in heat capacities (Fig. 5, Dong et al., 2009). Land near the Arctic generally warms the most and sees a slight increase in precipitation (Figs. 5, 6). The increased temperatures produce higher rates of evaporation for the heavier oxygen isotope and increases the residence time of water vapor in the atmosphere, which may contribute to an increased advective length scale of enriched moisture transport (Singh et al., 2016). Decreased relative humidity leads to decreased rates of precipitation - specifically in the subtropics at the dry, descending region of the Hadley Cell, which result in increased $\delta^{18}O_p$ in the 6x PI $CO_2$ simulations (Figs. 6, 7, A8, A17). There is also an increase in sea surface temperatures and evaporation over these subtropical warmer waters which populate the air mass with heavier oxygen isotopes and increase $\delta^{18}O_p$ (Figs. 7, A17).

Though most areas exhibit an increase in $\delta^{18}O_p$ with the 6x PI $CO_2$ modeled case, the western equatorial Pacific and some subpolar regions show a slight decrease (Fig. 7). These regions experience increases in relative humidity, cloud cover, and precipitation under the higher $CO_2$ background state, resulting in $^{18}O$-depleted rainfall (Figs. 6, A7, A8). The colder subpolar region does not experience much of an increase in surface air temperatures, ~2°C or less, which may have contributed to the slight increase, ~3% or less, in relative humidity (Figs. 5, A8). The equatorial Pacific sees an increase in temperature, but it also experiences highly increased rainfall (Fig. 6). There has been previous evidence of a narrowing and strengthening of the ITCZ during the onset of the PETM, which would increase precipitation, resulting in decreased $\delta^{18}O_p$ (Tierney et al., 2022; Cramwinckel et al., 2023; Byrne and Schneider, 2016). The narrowing tendency is largely due to the enhanced meridional moist static energy gradient seen in warming climates with increased atmospheric moisture (Fig. 6; Byrne and Schneider, 2016). Moreover, there is often a widening of the Hadley circulation projected in warming climates, which contributes to the expansion of a dry descent region off the tropics (Figs. 6, A17; Byrne and Schneider, 2016). With stronger, unsaturated downdrafts, there is an expected increase in $\delta^{18}O_p$ over most subtropical land (Fig. 7).

The North American Cordillera experiences large increases in $\delta^{18}O_p$ under the higher $CO_2$ conditions (Fig. 7). Areas of high elevation generally have very low $\delta^{18}O_p$ compared to areas of low elevation due to isotopic distillation during rainout. This elevation distinction is reduced during times of extremely warm temperatures due to atmospheric subsidence of vapor enriched

in $^{18}O$, which explains the substantial increase, up to 3‰, in $\delta^{18}O_p$ with a doubled $CO_2$ (Fig. 7, Poulsen and Jeffery, 2011). However, OrbMaxN does not see as dramatic an increase in $\delta^{18}O_p$ under higher $CO_2$ conditions at this mountain range (Fig. 7). The isotopic response is lower because the higher insolation in the NH at OrbMaxN already caused a response under the 3x PI $CO_2$ conditions (Fig. A13). $\delta^{18}O_p$ over the mountain range is higher in the OrbMaxN case than the other cases at 3x PI $CO_2$ because the increased insolation warmed the mountain range and increased $\delta^{18}O_p$ substantially, so there is a smaller difference in $\delta^{18}O_p$ between the $CO_2$ levels (Fig. 7).

Finally, most orbits display just a slight increase in $\delta^{18}O_p$ in northern Africa under the higher $CO_2$ state, but OrbMaxN experiences a substantial increase in $\delta^{18}O_p$ in northern Africa, up to 3‰, largely owing to drier conditions, including over the Indo-Pacific warm pool where most of the moisture is sourced (Figs. 6, 7, A8, A17). Northern Africa experiences a stronger decrease in relative humidity for OrbMaxN than any other orbit under the higher $CO_2$ state (Fig. A8). The decrease in relative humidity and precipitation results in more enriched rainfall causing a stronger increase in $\delta^{18}O_p$ over this desert and shrubland region under OrbMaxN conditions (Fig. 7). Additionally, the prevailing trade winds are also relatively cooler and drier over the Indo-Pacific in OrbMaxN, though the Indo-Pacific warm pool remains very warm under all orbits, resulting in a stronger temperature and moisture gradient at the air-sea interface. This gradient increases evaporation rates at the source, resulting in higher $\delta^{18}O_p$ over most of the Sahara Desert (Figs. 7, A8, A17).

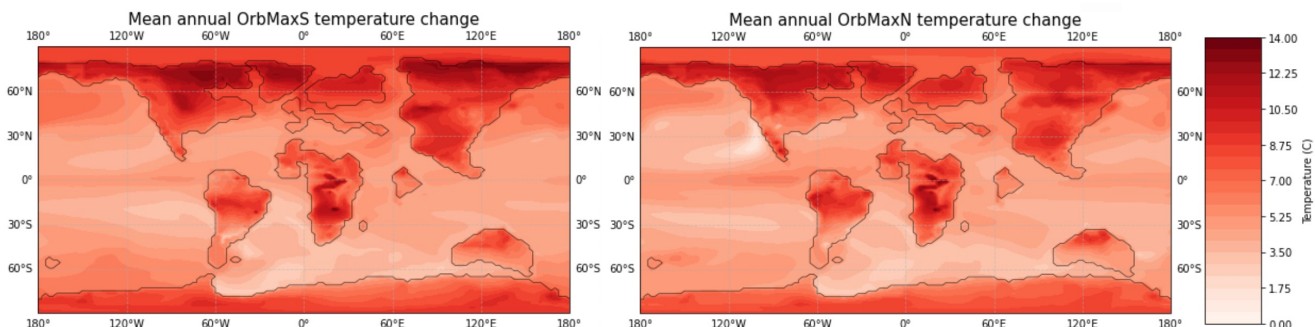

**Figure 5: The difference in mean annual surface temperatures at 6x PI $CO_2$ compared to 3x PI $CO_2$ for OrbMaxS (left) and OrbMaxN (right). To view absolute mean annual surface air temperatures, see Fig. A11.**

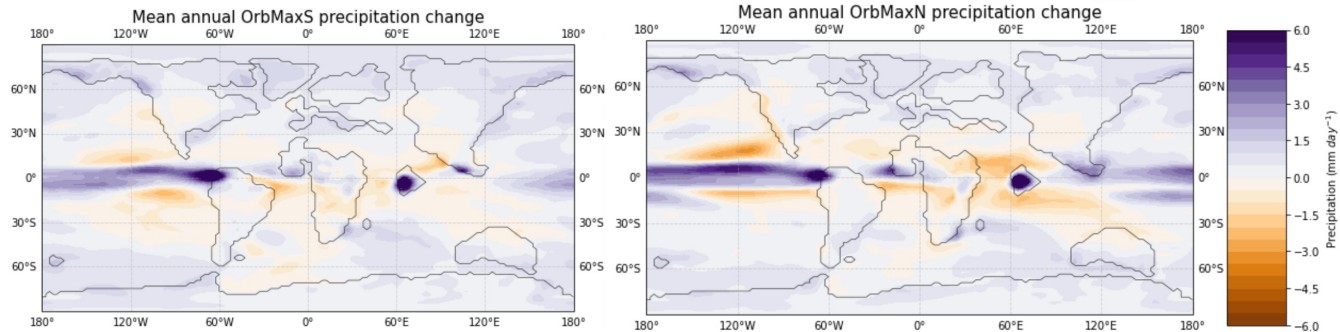

**Figure 6: The difference in mean annual precipitation under 6x PI CO₂ compared to 3x PI CO₂ for OrbMaxS (left) and OrbMaxN (right). The maximum positive precipitation difference on the corresponding color bar represents anything experiencing 6.0 mm day⁻¹ or higher. To view absolute mean annual precipitation, see Fig. A12.**

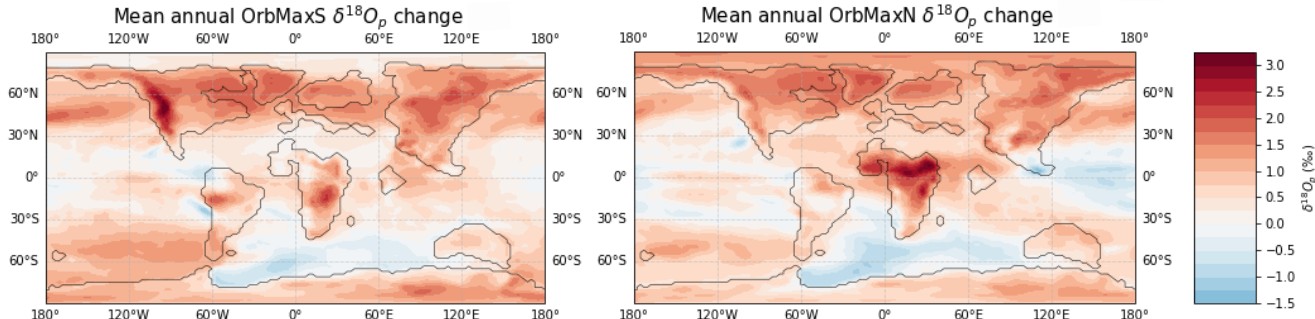

**Figure 7: The difference in mean annual $\delta^{18}O_p$ at 6x PI CO₂ compared to 3x PI CO₂ for OrbMaxS (left) and OrbMaxN (right). To view absolute mean annual $\delta^{18}O_p$, see Fig. A13.**

### 3.3 Orbital sensitivity under different CO₂ background states

Given the above results, the hydrologic cycle is clearly impacted by both changes in orbit and changes in atmospheric CO₂ levels. However, none of the above results address the potential difference in the hydrologic cycle's orbital sensitivity under two different CO₂ background states. In other words, do changes in orbit at a lower CO₂ level have a greater impact on the oxygen isotopic ratio of precipitation than changes in orbit at a higher CO₂ level? When comparing the $\delta^{18}O_p$ differences between OrbMaxS and OrbMaxN at 3x PI CO₂ versus 6x PI CO₂, the spatial patterns of enrichment and depletion of heavy oxygen isotopes are similar (Figs. 4, 8). However, the global mean annual $\delta^{18}O_p$ difference between these two orbits is over 20% smaller for 6x PI CO₂ than 3x PI CO₂. Averaged globally, the change in orbit has a smaller impact on the hydrologic cycle at the higher CO₂ level, representative of the PETM.

Additionally, the regions that experience a larger enrichment or depletion of heavy oxygen isotopes between orbits during certain seasons, like western North America or northern Africa, see a much smaller $\delta^{18}O_p$ difference at the higher CO₂ level than the lower CO₂ level, by as much as 4‰ (Figs. 4, 8). Therefore, the higher atmospheric CO₂ level dampens the orbital

sensitivity of the hydrologic cycle, especially in these regions. Fractionation of oxygen isotopes is more pronounced at lower temperatures, and the 3x PI $CO_2$ background state exhibits much lower global temperatures than the 6x PI $CO_2$ background state (Fig. 5, Luz et al., 2009). $CO_2$-induced warming tends to slow general circulation in the tropics and subtropics (Singh et al., 2016). Higher temperatures result in higher rates of evaporation, an increase in water vapor residence time, and more $^{18}O$

in the atmosphere, which causes a lower fractionation factor between the lighter and heavier oxygen isotopes and less rainout and distillation (Luz et al., 2009). The smaller fractionation rate between oxygen isotopes during evaporation and decrease in rainout and distillation results in smaller, muted differences in the oxygen isotope variability of precipitation between orbits. Therefore, the change in insolation distribution has less of a considerable impact on the hydrologic cycle at higher atmospheric $CO_2$ levels.


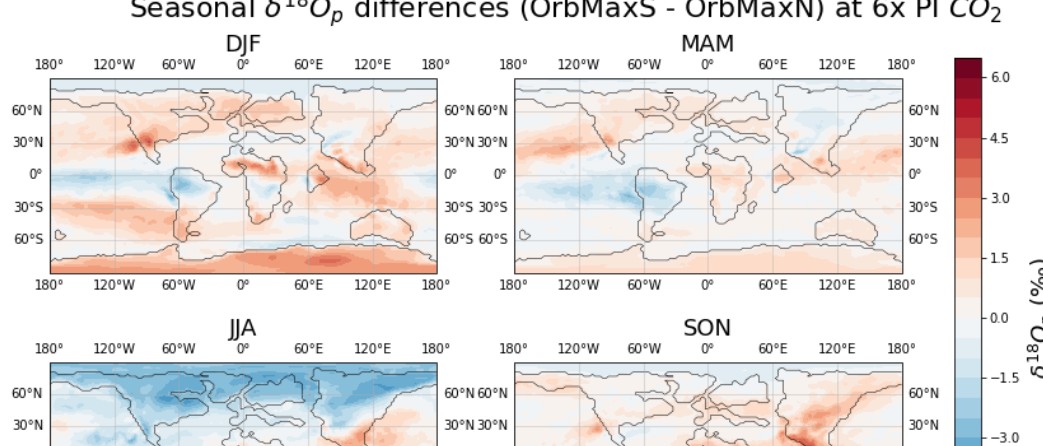

**Figure 8: The seasonal $\delta^{18}O_p$ differences between OrbMaxS and OrbMaxN at 6x PI $CO_2$.**

**3.4 Model-data comparison**


Isotope-enabled climate models can simulate the transformation and transportation of water isotopes in all components of the model, which allows for direct comparison between modeled isotopic ratios and paleo-recorded isotopic ratios, and further assessment of uncertainties (Zhu et al., 2017). This study gathers $\delta^{18}O$ data from soil carbonates and $\delta^2H$ data from leaf waxes in order to validate iCESM's simulated terrestrial hydrologic cycle during the PETM, which had more available data than the

Early Eocene. The lower $CO_2$ background state, representative of the greater Early Eocene age, is validated through a terrestrial temperature model-data comparison (Fig. A2, Zhu et al., 2019). The isotopic model-data comparison results focus only on the

6x PI $CO_2$ simulations and PETM records. The comparisons account for seasonality by including both the highest and lowest mean monthly soil water $\delta^{18}O$ or $\delta^2H$ (dashed lines) for all terrestrial longitudes at the given latitude for each orbital simulation, along with the mean annual values (solid lines) (Fig. 9, Fig. 10). These figures concisely capture the entire seasonal range of

simulated isotopic signals latitudinally.  To account for regional effects rather than global effects, we also produce a point-by-point comparison of proxy isotopic data with simulated isotopic data at the grid cells in the model that correspond with the paleo-coordinates of each proxy record (Figs. A14, A15). These figures include mean annual and mean summer isotopic data to investigate potential seasonal biases in the data, as well as biases within the model.

About 60% of PETM soil carbonate records fall within the simulated seasonal range for soil water $\delta^{18}O$, and several siderite records fall above the highest monthly means, which likely represent the warm season (Fig. 9). In order to capture as many proxy records as possible and not assume a particular seasonal time of formation, Fig. 9 displays the largest possible range by utilizing the lowest and highest monthly modeled soil water $\delta^{18}O$ at each latitude. However, we also created the same figure using specifically summer and winter means, though the range becomes tighter (Fig. A9). The siderite record may be warm

season biased, and the model may exhibit a slightly low $\delta^{18}O$ bias. Further assessment of model and proxy biases contributing to this occurrence can be found in Sect. 4.

About 70% of PETM leaf wax records fall within the simulated seasonal range for model-inferred leaf wax $\delta^2H$ (Fig. 10). In order to capture as many proxy records as possible and not assume a particular seasonal time of formation, Fig. 10 displays

the largest possible range by utilizing the lowest and highest monthly model-inferred leaf wax $\delta^2H$ at each latitude. However, we also created the same figure using specifically summer and winter means, though the range becomes tighter (Fig. A10). Records outside of the largest range may be seasonally biased or have a different fractionation factor than the model-inferred values. Further assessment of model and proxy biases contributing to this occurrence can be found in Sect. 4.

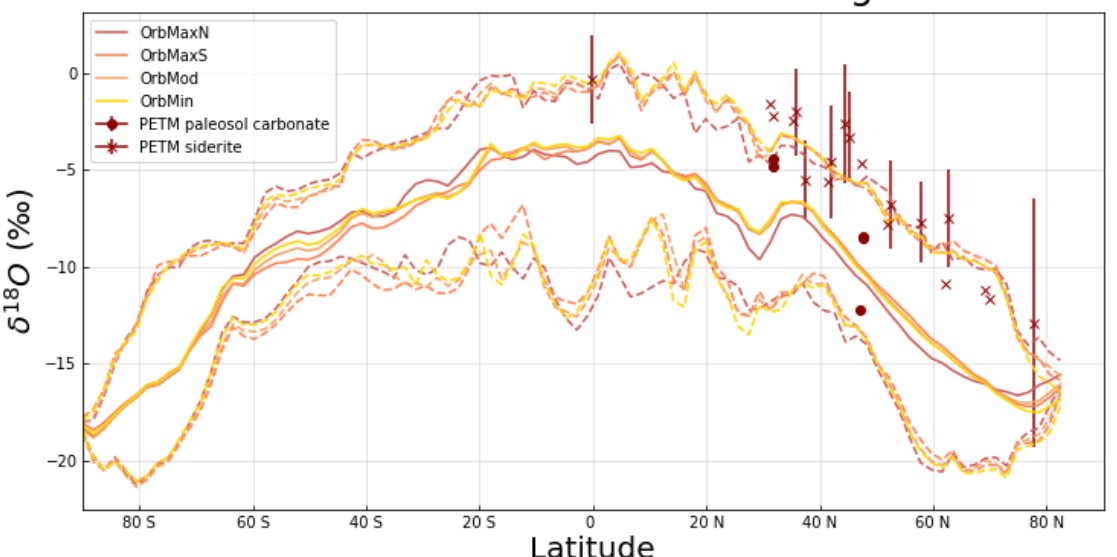


**Figure 9: The simulated seasonal range of soil water $\delta^{18}O$ at a depth of 40-100 cm for each orbit under 6x PI CO₂ conditions compared to PETM paleosol carbonate and siderite $\delta^{18}O$ records. The middle solid lines represent the mean annual soil water $\delta^{18}O$ at each latitude for all terrestrial longitudes, and the lower and upper dashed lines represent the lowest and highest monthly means at each latitude, respectively. The error bars represent uncertainty. The specific summer and winter means can be found in Figure**

**A9.**

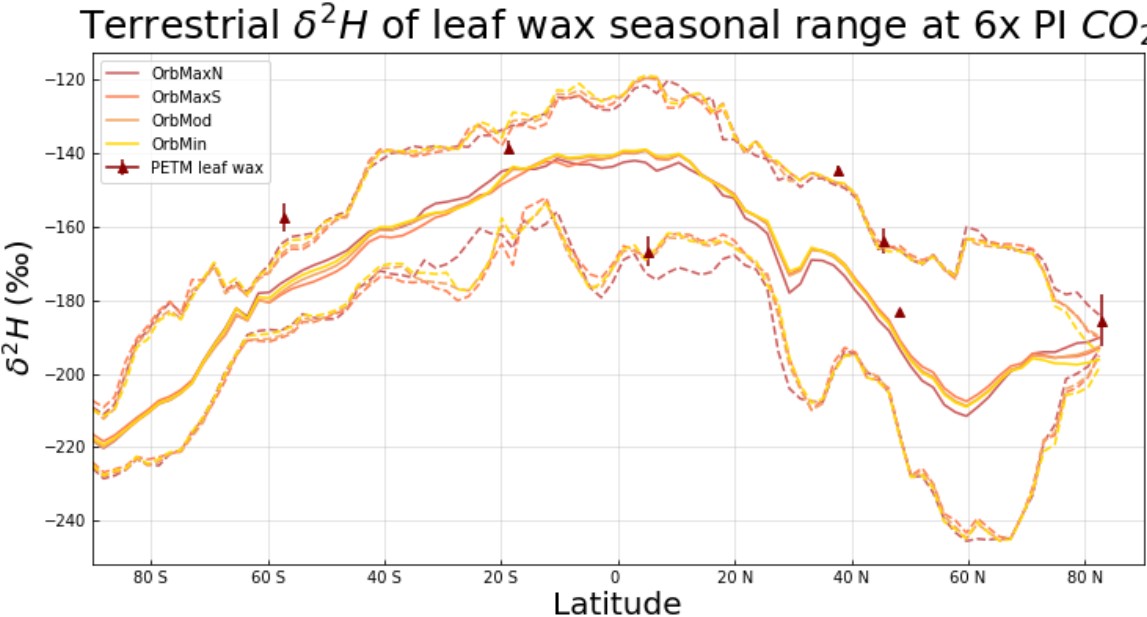

**Figure 10: The seasonal range of model-inferred leaf wax $\delta^2H$ for each orbit under 6x PI CO₂ conditions compared to PETM leaf wax $\delta^2H$ records. The middle solid lines represent the mean annual leaf wax $\delta^2H$ at each latitude for all terrestrial longitudes, and**

**the lower and upper dashed lines represent the lowest and highest monthly means at each latitude, respectively. The error bars represent uncertainty. The specific summer and winter means can be found in Fig. A10.**

The PETM was a relatively short period of extreme warming, and the eccentricity configuration would have remained almost constant during the onset. Furthermore, orbit has a substantial influence on climate, so a best guess as to the orbit during the
onset of the PETM would impact our understanding of the PETM global warming and how it compares to present-day global warming. Figures 9 and 10 fail to emphasize orbital differences, as they only display zonal seasonal range, so this study employs a quantitative point-by-point comparison method to emphasize orbital differences. The simulated 6x PI CO$_2$ $\delta^{18}$O$_p$ values are compared to the Van Dijk et al (2020) siderite PETM record to evaluate which orbital simulation is the best match. This record includes the widest span of latitudes and longitudes and analyzed several samples at each location to find a mean
$\delta^{18}$O value. In order to practically compare these proxy values to the simulated values, we found which group of 4 model grid cells captured the paleo-coordinates of each proxy location, and then found the NH summer mean $\delta^{18}$O values for that group of cells for each simulation. The siderite proxy values originate from the NH and appear to represent summer values (Table A1, Figs. 9, A14). The modeled summer days have been adjusted for the paleo-calendar effect, which structures time as a fixed number of degrees in Earth's orbit, rather than a fixed number of days each month, so that seasonal comparisons between
simulation and data are properly lined up according to the Earth's position in its orbit (Bartlein and Shafer, 2019). The simulated $\delta^{18}$O summer means at each of the ten locations were then used to calculate the root mean square deviation (RMSD), calculated following Eq. (1):

$$RMSD = \sqrt{\frac{\sum_{i=1}^{N}(\chi_i - \hat{\chi}_i)^2}{N}} \, . \tag{1}$$

RMSD is commonly used for model-data comparisons (Flato et al., 2013; Thompson et al., 2022). The lower the result, the more comparable the simulated values are to the proxy values. The 6x PI CO$_2$ and OrbMaxN run is in best alignment with the proxy record. However, we are limited by the lack of SH Van Dijk records representing the PETM. The Van Dijk RMSD values are in bold (Table 3). We also decided to complete the RMSD with the other $\delta^{18}$O records (first number in parentheses), as well as the $\delta^2$H records (second number in parentheses), to see if their pattern of comparability would be consistent with the
Van Dijk siderite record. All proxy records are more comparable to the 6x PI CO$_2$ simulations than the 3x PI CO$_2$ simulations since they are all from the PETM, but the orbital preferences varied (Table 3). Other hesitations, drawbacks, and speculations can be found in Sect. 4.

| Root Mean Square Deviation | 3x PI CO$_2$ | 6x PI CO$_2$ |
|---|---|---|
| OrbMaxN | **3.010** (3.426, 15.804) | **2.652** (2.924, 14.263) |
| OrbMaxS | **4.932** (3.354, 14.457) | **3.323** (2.262, 13.924) |
| OrbMin | **4.338** (3.196, 15.667) | **3.059** (2.436, 14.436) |

| OrbMod | **4.260** (3.298, 15.517) | **3.149** (2.437, 14.288) |
| --- | --- | --- |

Table 3: The calculated RMSD for each simulated case compared to the Van Dijk siderite record (in bold). The parentheses hold the calculated RMSD for each simulated case compared to the other $\delta^{18}O$ records, as well as the $\delta^2H$ records, respectively.

## 4 Discussion

These Early Eocene sensitivity experiments provide a window to large-scale climate patterns and paint some environmental context for proxy records. Our model simulated four orbits at two different atmospheric $CO_2$ settings and traced water isotopic ratios to allow for comparisons to various water isotope proxy records. A past study using a lower resolution model of the Early Eocene found a heavy influence of orbit on seasonal precipitation trends during the Early Eocene and our findings agree (Keery et al., 2018). OrbMaxS and OrbMaxN simulations exhibit high eccentricity and obliquity, resulting in intense seasonality, but different orbital precession, resulting in dramatic differences in insolation distribution seasonally (Fig. A3).

We find dramatic differences in $\delta^{18}O_p$ between OrbMaxS and OrbMaxN as a result of seasonal changes in insolation, reaching up to ~6.5‰ in certain regions (Fig. 4). The differences in $\delta^{18}O_p$ between the two $CO_2$ background states were less dramatic, reaching only ~3.0‰ in certain regions (Fig. 7). Key regions of interest (those experiencing the greatest differences in oxygen isotopic signals) show greater changes in $\delta^{18}O_p$ in response to orbital change than a $CO_2$ doubling due to higher seasonal sensitivity. This may be key to explaining some variability within terrestrial water isotope paleo-records. For instance, the variation in the terrestrial $\delta^2H$ leaf record in Inglis et al (2022) may be partly attributed to orbital variability. However, the simulations with doubled $CO_2$ showed a consistent, mean annual increase in $\delta^{18}O_p$, which orbital changes are less likely to provoke. Therefore, changes in precession play an important role on the hydrologic cycle seasonally and is a valuable piece of information to consider when interpreting paleo-records – especially when those records form over shorter periods of time and are seasonally biased.

Additionally, we find that orbital variability has relatively greater influence on precipitation isotopes under a lower $CO_2$ condition. This may imply that orbit exerts more control on the seasonal hydrological cycle in colder climates than warmer climates. As such, it may be especially important to incorporate the potential influence of orbital variability on colder, long-interval climate studies in future work. Studying orbital control on the hydrological cycle in warmer climates is still recommended, but it may have slightly less considerable of an impact in extremely high $CO_2$ environments.

Some paleosol carbonate proxies most closely match maximum simulated soil water $\delta^{18}O$, which represent summer values, though not all (Fig. 9). This comparison is consistent with the idea that paleosol carbonates sometimes preserve a signal of the isotopic composition of rainfall during the warmer, more evaporative season in which the carbonates may be more likely to form (Kelson et al., 2020). The siderite records appear to fall along the maximum $\delta^{18}O$ simulated values for all modeled cases,

signaling a possible warm-season bias. Recent studies argue that pedogenic siderite forms between the mean annual air temperature and the mean air temperature of the warmest months, depending on the latitude (Fernandez et al., 2014; van Dijk et al., 2019). Some temperature reconstructions support that siderite forms more rapidly under warmer, more evaporitic

conditions, especially in higher latitudes, since it's controlled by microbial iron reduction which proceeds faster in higher soil temperatures (van Dijk et al., 2020). Therefore, both archives likely have some records that represent the soil water $\delta^{18}O$ during the warm, dry, evaporative season when they are more likely to form, but especially the siderite. As siderite appeared consistently warm season biased, we chose to use the summertime average soil temperature for the siderite PSM. The summer season receives the greatest insolation, which increases temperature and evaporation rates, which in turn would have biased

the isotope recording if this environment did encourage faster soil carbonate growth. This bias is seen in the point-by-point comparison as well, which highlights regional climate over global climate, since the simulated mean summer isotopic signals more closely mimic the proxy data than the simulated mean annual isotopic signals (Figs. A14, A15).

The terrestrial $\delta^{18}O$ proxy records span much of the NH, but are lacking in the SH, limiting our model-data comparison.

Although the model's paleo-elevation roughly matches the paleo-elevation estimates from the proxy records, proxies from the highest elevations were excluded because paleo-altimetry estimates have larger uncertainty. Aside from the seasonal bias, the exclusion of high elevation proxies may explain why none of the records sit closer to the minimum $\delta^{18}O$ values, as areas of high elevation often result in low $\delta^{18}O$, though this is not necessarily the case under high $CO_2$ conditions (Dansgaard, 1964). Additionally, there is evidence that iCESM1.2 has a slight low bias for $\delta^{18}O_p$, which may transfer to the $\delta^{18}O$ of soil water and

drive further misalignment between model and proxies (Brady et al., 2019). This slight low bias may partially explain the misalignment between simulated and proxy isotopic signals that is seen in the point-by-point comparison as well (Figs. A14, 15). The model also exhibits some depleted bias in $\delta^2H$, as well as the presence of a double ITCZ, which may encourage biases in extratropical moisture transport (Brady et al., 2019). Other limitations in this model-data comparison may include the possibility of evaporation before burial, diagenesis, uncertainty in timing, varying elevations, or error in paleo-coordinate

conversion or fractionation factor.

Furthermore, the terrestrial $\delta^2H$ proxy records largely fall within the simulated seasonal range for model-inferred leaf wax $\delta^2H$ (Fig. 10). The Jaramillo et al (2010) record is closer to the minimum $\delta^2H$, likely because that record was taken from a tropical rainforest with increased and $^2H$-depleted rainfall, but the other records align closer to mean or maximum $\delta^2H$ (Fig. 10,

Jaramillo et al., 2010). Leaves undergo transpiration while growing, an evaporative process that further fractionates water isotopes, which often result in $^2H$-enrichment of leaf water and has a critical effect on the final $\delta^2H$ of leaf wax n-alkanes (Kahmen et al., 2013). Therefore, it is likely that some records align closer to maximum $\delta^2H$ values in part due to transpiration. Leaves also tend to grow over a matter of weeks, so seasonal bias and short-term formation could be another reason for slight mismatches between the model and data. Plus, the fractionation factor used in the WaxPSM is a globally averaged estimate,

and there's a wide range of potentially realistic fractionation factors that could shift leaf wax $\delta^2H$ values by as much as ~20‰

(Handley et al., 2012; Pagani et al., 2006). The Earth experienced a major precession-driven modification of global vegetation during the PETM and across the Eocene, so the changes in biosphere-atmosphere interactions and plant biology could have significantly impacted the hydrological cycle and leaf wax isotopes (Tardif et al., 2021). Aside from that, some of the same limitations faced in the $\delta^{18}O$ comparison are also relevant in the $\delta^2H$ comparison – diagenesis, uncertainty in timing, varying elevations, or error in paleo-coordinate conversion. However, the majority of leaf wax records fall within the model-inferred seasonal range of leaf wax $\delta^2H$.

Finally, previous studies indicated that Earth experienced near-maximum solar irradiance at the onset of the PETM largely due to variations in eccentricity (Zeebe et al., 2017; Lourens et al., 2005; Westerhold et al., 2009; Zachos et al., 2010; Galeotti et al., 2010; Westerhold et al., 2012). Several studies argue that the Earth's eccentricity at this time may have partly caused the extreme warming during the PETM, and our findings agree (Kiehl et al., 2018; Lawrence et al., 2003). The Kiehl et al (2018) study also argues that the Earth was likely experiencing an orbit most similar to OrbMaxN at the onset of the PETM (Table 3). Although it is worth constraining the orbit at the onset of the PETM in order to further understand the relatively rapid and extreme warming that followed, there are several limitations to this model-data comparison that render it less effective. There are biases with the oxygen isotope records, discussed above, and several drawbacks of the simulations, including the resolution and model bias. These model simulations are run with a relatively coarse atmosphere (~2-degree horizontal resolution) and topography, which may not fully capture the local environments of the proxy records. Perhaps most importantly, the timing of the onset of the PETM is not perfectly constrained so the proxy records may not represent the onset itself. So aside from the van Dijk siderite record, we also conducted an RMSD for the other PETM proxy records to speculate on what consistent (or inconsistent) patterns between simulations could mean. All records match the higher $CO_2$ level better than the lower $CO_2$ level within the same orbit, which was expected with PETM records (Table 3). However, the other $\delta^{18}O$ records did not match OrbMaxN best, but rather matched OrbMaxS best at the PETM greenhouse gas level. The $\delta^2H$ records seem to favor OrbMaxS as well, if anything. Given the variability within the RMSD, we hesitate to draw any strong conclusions, but we speculate that the records all likely formed during varying orbits or were slightly more likely to form during orbital times of maximum seasonality.

Determining the potential orbit in existence at this time can contribute to our knowledge of how this past global warming differs from our present-day global warming. For instance, the 6x PI $CO_2$ and OrbMaxN modeled run is relatively far from our present-day warming scenario. The current atmospheric $CO_2$ concentration is about ¼ as high, and the 6x PI $CO_2$ OrbMaxN simulation has a mean annual global temperature 0.71°C higher than the 6x PI $CO_2$ OrbMod simulation, largely driven by the higher insolation maximum of OrbMaxN. If the 6x PI $CO_2$ OrbMaxN simulation is the closest to representing the onset of the PETM, this highlights how the PETM differs from modern climate in important ways that constrain its use as an analogue for the Anthropocene, especially since a maximum NH summer insolation orbit would have slightly bolstered global warming during the PETM, unlike our modern orbit.

## 5 Conclusions

This study demonstrates the relative impact of $CO_2$ and orbit on climate and the orbital sensitivity of the hydrologic cycle under different $CO_2$ background states through the combination of a fully coupled climate model suite of the Early Eocene and published terrestrial $\delta^{18}O$ and $\delta^2H$ proxy records. Our results reveal that large variations in orbit can have a more substantial impact on the hydrologic cycling than doubling $CO_2$ in certain regions seasonally, although doubling $CO_2$ has a more consistent and global-scale impact, and that orbital sensitivity weakens under a higher $CO_2$ background state. These findings highlight the importance of modeling various orbital states to understand variation in water isotope records and stress the influence changes in orbit have on seasonal climate relative to changes in greenhouse gases. This study also determines that the concentration of greenhouse gases in the atmosphere partly controls the sensitivity of the climate to orbital changes. The comparison to terrestrial $\delta^{18}O$ and $\delta^2H$ records verifies the reliability of our model and lends some interpretation to the biases within iCESM and the environmental context pertaining to the potential warm-season bias of some of the proxies so we may better understand what they represent. The iCESM generally performs well in simulating hydrologic cycling during a warmer climate, which increases trust in iCESM to project future climate change, though some water isotope biases exist within the realm of the model. The empirical evidence appears to host biases as well, especially siderite with a seasonal bias, which underscores the importance of taking orbit into account when assessing siderite records. Exploring OrbMaxN as a potential cause for the onset of the PETM is worthwhile, though many uncertainties in the model-data comparison exist. Simulating the Early Eocene is valuable as it experienced high $CO_2$ levels, heightened seasonality, and precipitation extremes, which we expect with future climate change. These simulations further our understanding of the role $CO_2$ and orbit play in climate change and the hydrologic cycle.

## Appendix A

| Modern Coordinates (°N, °E) | Paleo Coordinates (°N, °E) | Grid Cells (Lat, Lon) | Site Name |
|---|---|---|---|
| 62, -6.5 | 57.88, -10.4 | 78-79, 139-140 | Faroe |
| 33.4, -117.2 | 37.36, -99.78 | 67-68, 104-105 | Alberhill |
| 75.3, 135.5 | 77.69, 128.26 | 88-89, 51-52 | Arctic |
| 34.55, -92.51 | 35.8, -74.96 | 66-67, 114-115 | Arkansas |
| 47.32, -122 | 52.49, -102.24 | 75-76, 103-104 | Blum |
| 56.33, -133.22 | 62.66, -111.69 | 80-81, 99-100 | Alaska |
| 4, -72 | -0.24, -58.7 | 47-48, 120-121 | Colombia2 |
| 39.02, -104.27 | 41.92, -85.8 | 69-70, 109-110 | Colorado |

| 41.99, -106.44 | 45.15, -87.26 | 71-72, 109-110 | Hanna |
| 49, 2.2 | 44.3, -0.95 | 70-71, 142-143 | France |

**Table A1: The modern and paleo-coordinates for each van Dijk (2020) proxy location, followed by the appropriate model grid cells and site names.**

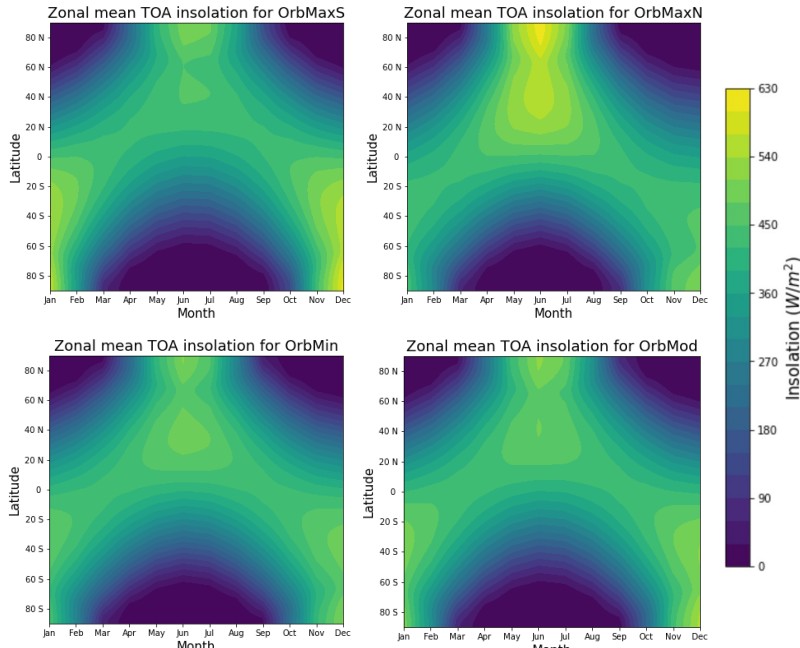

**Figure A1: The zonal, monthly top-of-atmosphere (TOA) solar insolation distribution for all orbits. OrbMaxS and OrbMaxN reach a higher summer insolation than OrbMod or OrbMin.**

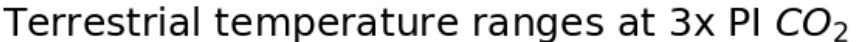

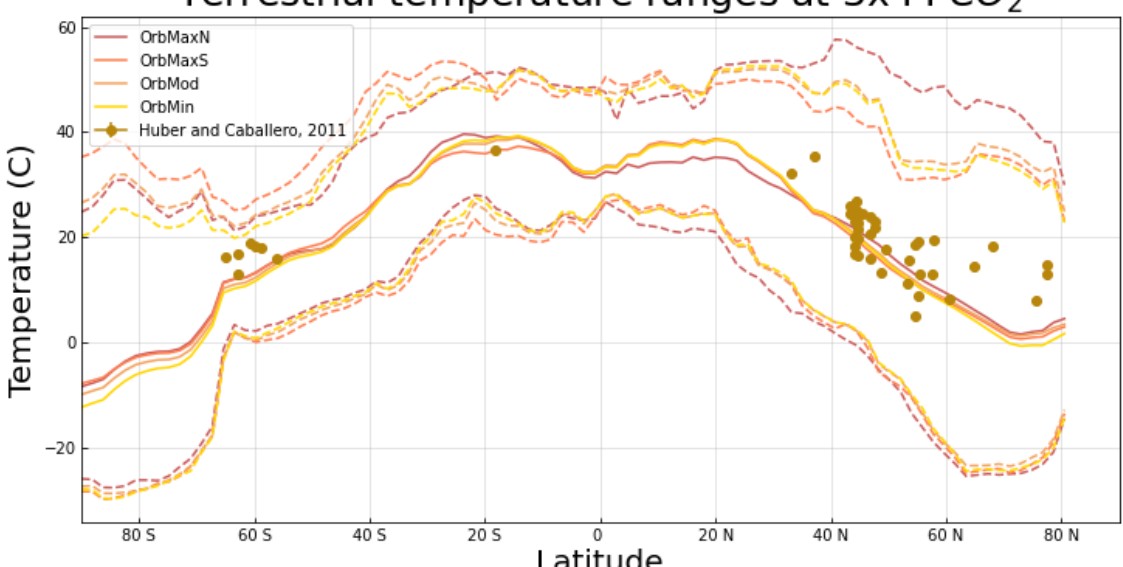

**Figure A2: Model-data comparison between the Huber & Caballero (2011) Eocene terrestrial dataset and the 3x PI CO$_2$ simulations. The middle solid lines represent the mean annual at each latitude for all terrestrial longitudes, and the lower and upper dashed lines represent the lowest and highest monthly means at each latitude, respectively.**

555

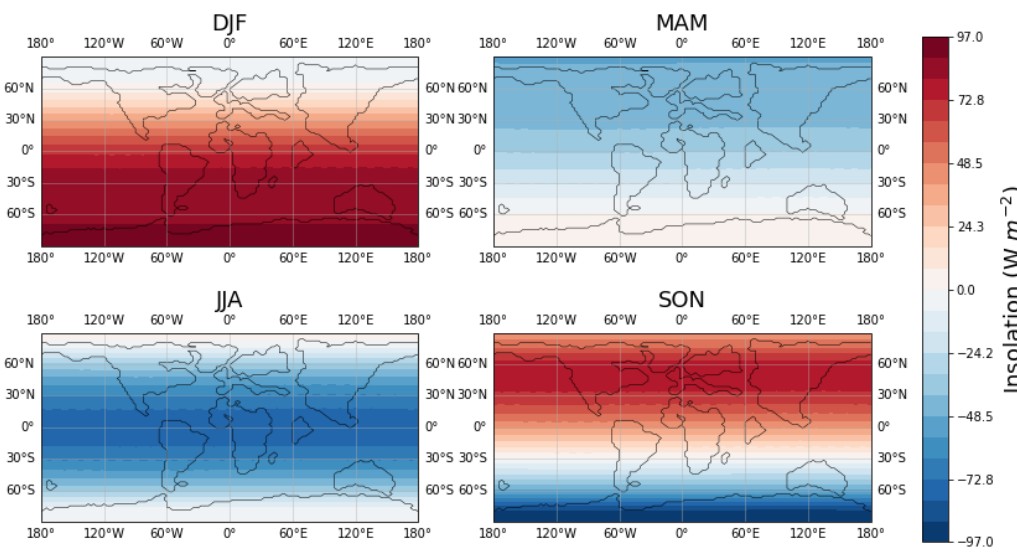

**Figure A3: The seasonal zonal top-of-atmosphere (TOA) solar insolation difference between OrbMaxS and OrbMaxN.**

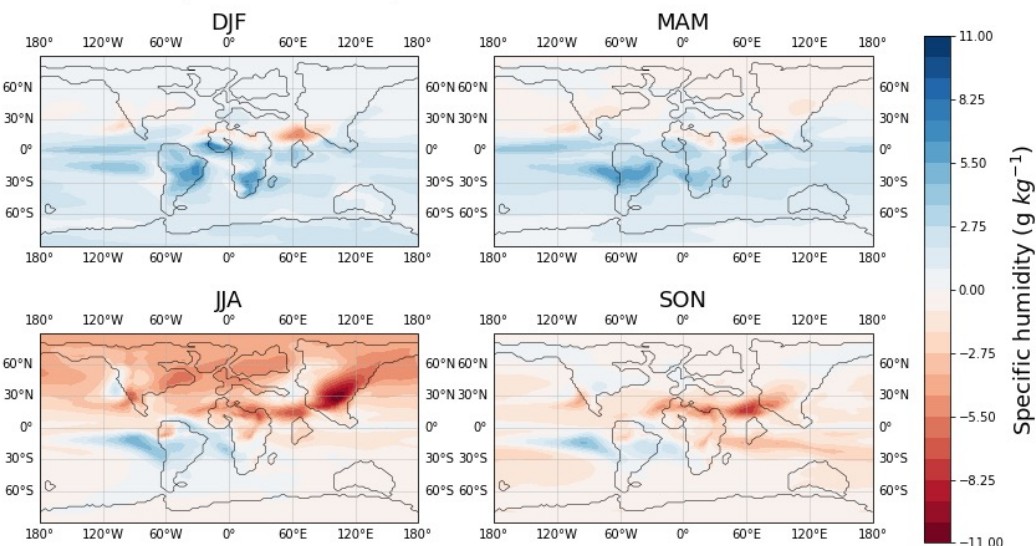

Figure A4: The seasonal specific humidity differences between OrbMaxS and OrbMaxN at 3x PI $CO_2$. Specific humidity is presented here at 850 atmosphere hybrid sigma pressure coordinates. Sigma coordinates represent pressure at the Earth's surface rather than the mean sea level, so it follows the actual terrain. We chose these coordinates because they represent the same pressure level above the land without running into mountains.

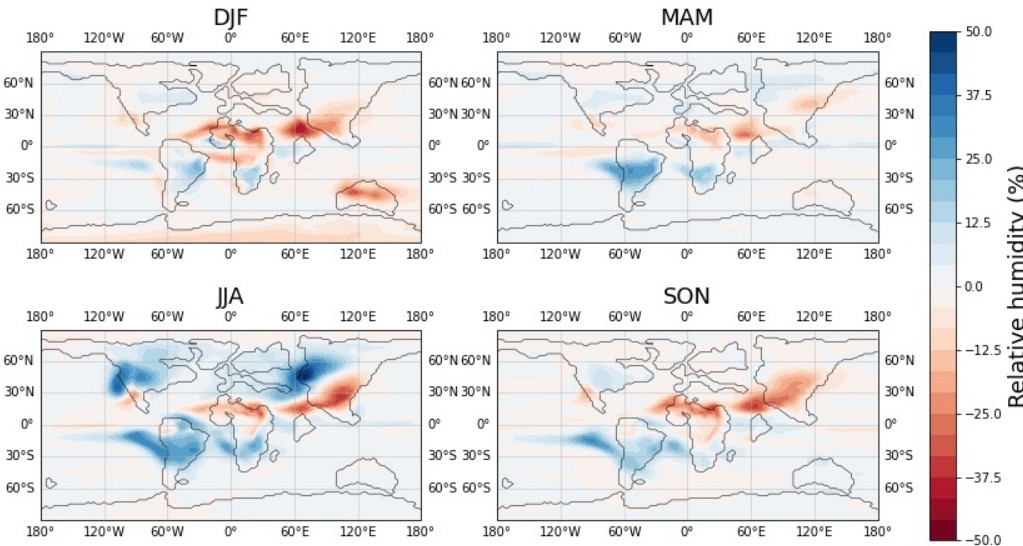

Figure A5: The seasonal relative humidity differences between OrbMaxS and OrbMaxN at 3x PI $CO_2$. Relative humidity is presented here at 850 atmosphere hybrid sigma pressure coordinates. Sigma coordinates represent pressure at the Earth's surface rather than the mean sea level, so it follows the actual terrain. We chose these coordinates because they represent the same pressure level above the land without running into mountains.

## Seasonal $\delta^2H_p$ differences (OrbMaxS - OrbMaxN) at 3x PI $CO_2$

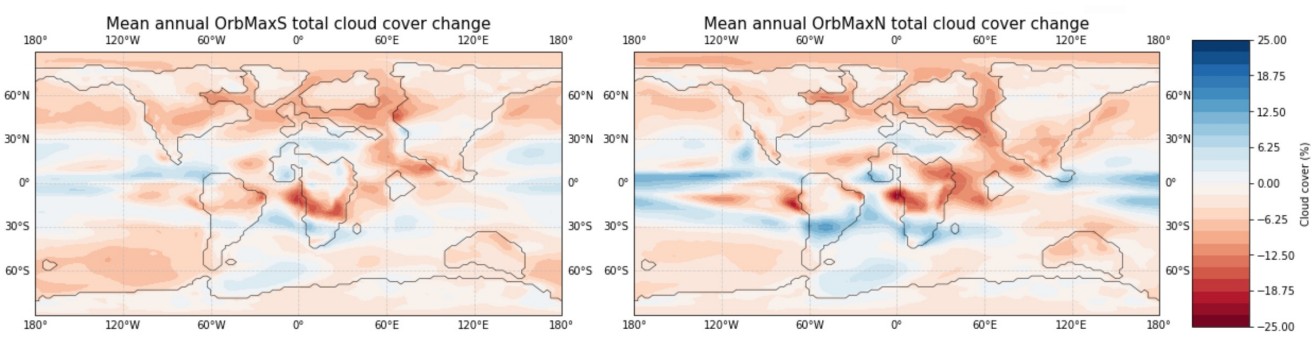

**Figure A6: The seasonal $\delta^2H_p$ differences between OrbMaxS and OrbMaxN at 3x PI $CO_2$.**

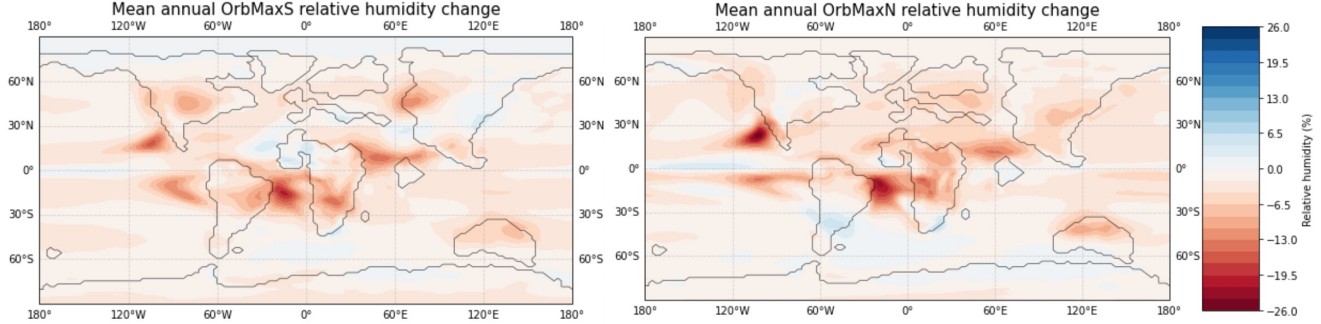

 **Figure A7: The difference in mean annual total cloud coverage under 6x PI $CO_2$ compared to 3x PI $CO_2$ for OrbMaxS (left) and OrbMaxN (right).**

**Figure A8: The difference in mean annual relative humidity at 6x PI $CO_2$ compared to 3x PI $CO_2$ for OrbMaxS (left) and OrbMaxN**
 **(right). Relative humidity is presented here at 850 atmosphere hybrid sigma pressure coordinates. Sigma coordinates represent**

pressure at the Earth's surface rather than the mean sea level, so it follows the actual terrain. We chose these coordinates because they represent the same pressure level above the land without running into mountains.

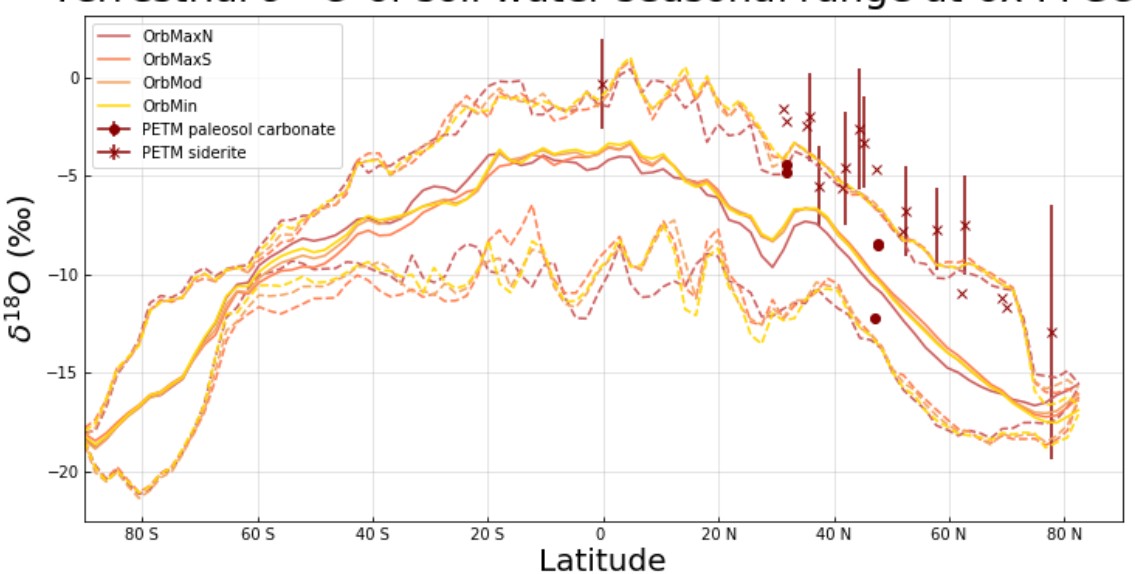

585   Figure A9: The simulated seasonal range of soil water $\delta^{18}O$ at a depth of 40-100 cm for each orbit under 6x PI $CO_2$ conditions compared to PETM paleosol carbonate and siderite $\delta^{18}O$ records. The middle solid lines represent the mean annual soil water $\delta^{18}O$ at each latitude for all terrestrial longitudes, and the lower and upper dashed lines represent the winter and summer means at each latitude, respectively. The error bars represent uncertainty.

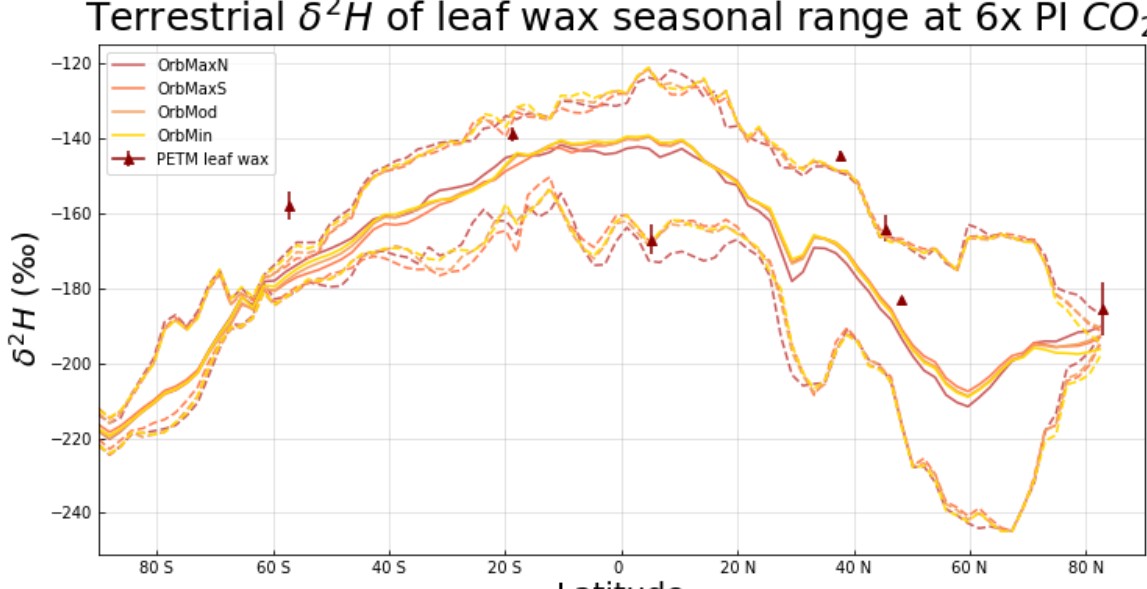

590

**Figure A10: The seasonal range of model-inferred leaf wax $\delta^2$H for each orbit under 6x PI $CO_2$ conditions compared to PETM leaf wax $\delta^2$H records. The middle solid lines represent the mean annual leaf wax $\delta^2$H at each latitude for all terrestrial longitudes, and the lower and upper dashed lines represent the winter and summer means at each latitude, respectively. The error bars represent uncertainty.**

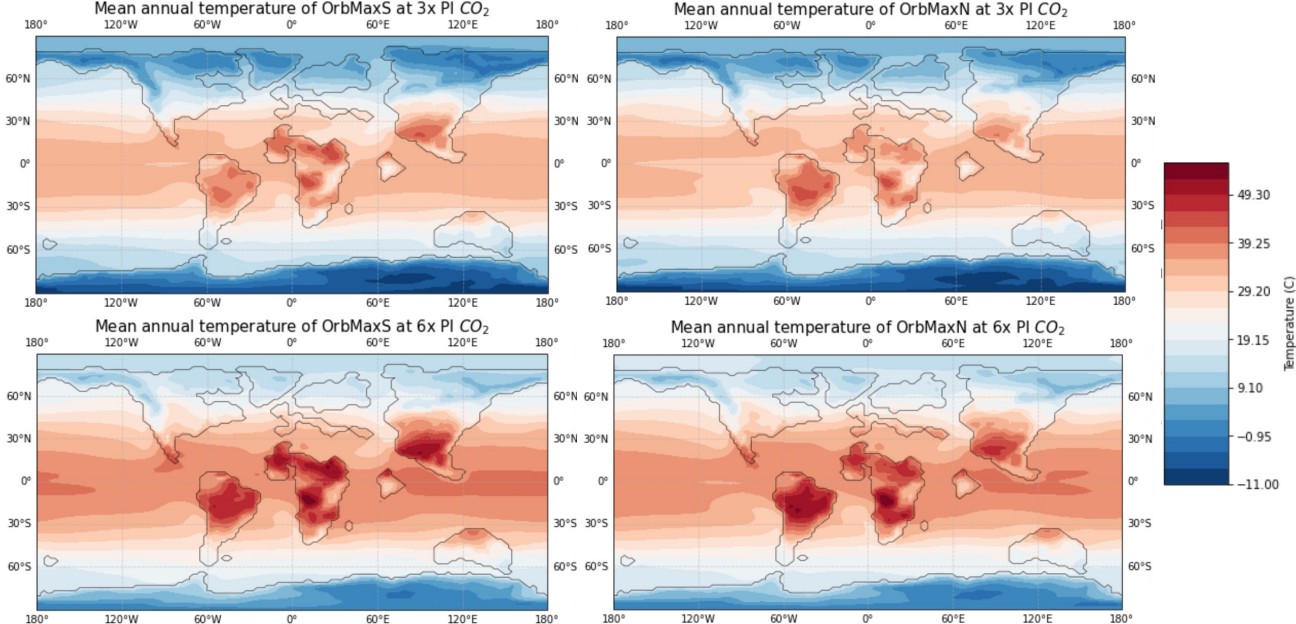

**Figure A11: The mean annual surface air temperature at 3x PI $CO_2$ (above) and 6x PI $CO_2$ (below) for OrbMaxS (left) and OrbMaxN (right).**

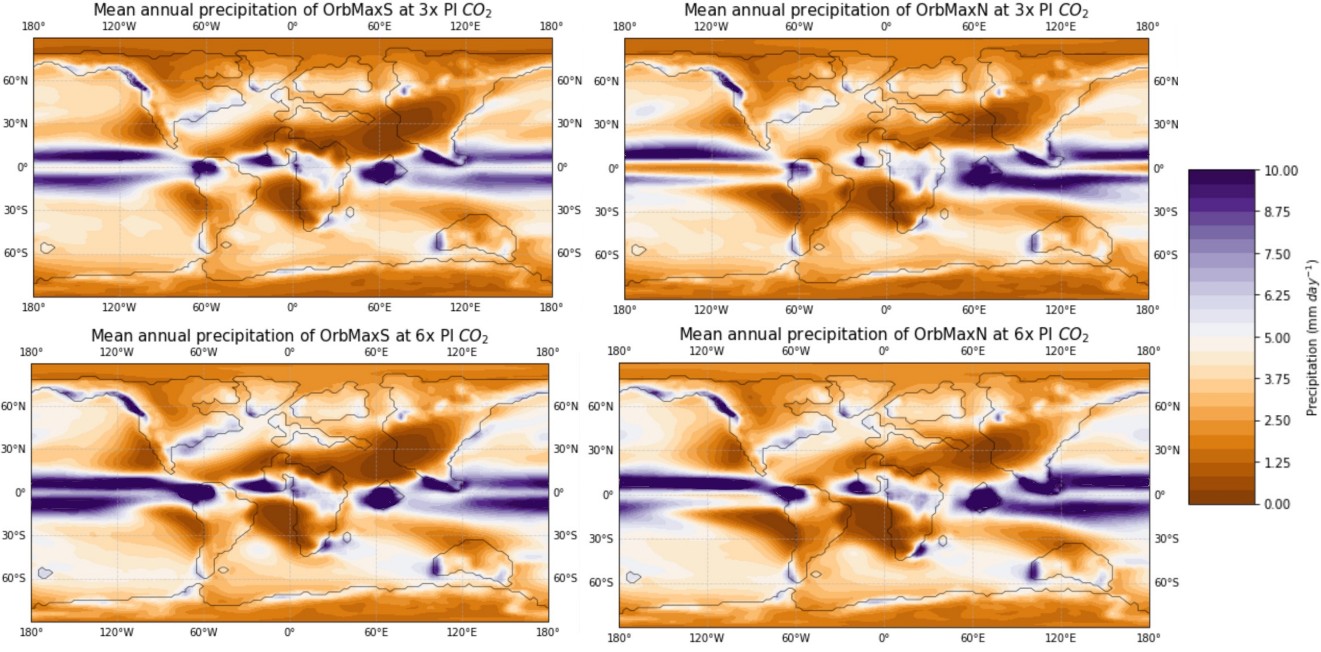

**Figure A12: The mean annual precipitation at 3x PI CO$_2$ (above) and 6x PI CO$_2$ (below) for OrbMaxS (left) and OrbMaxN (right). The maximum positive precipitation difference on the corresponding color bar represents anything experiencing 10.00 mm day$^{-1}$ or higher.**

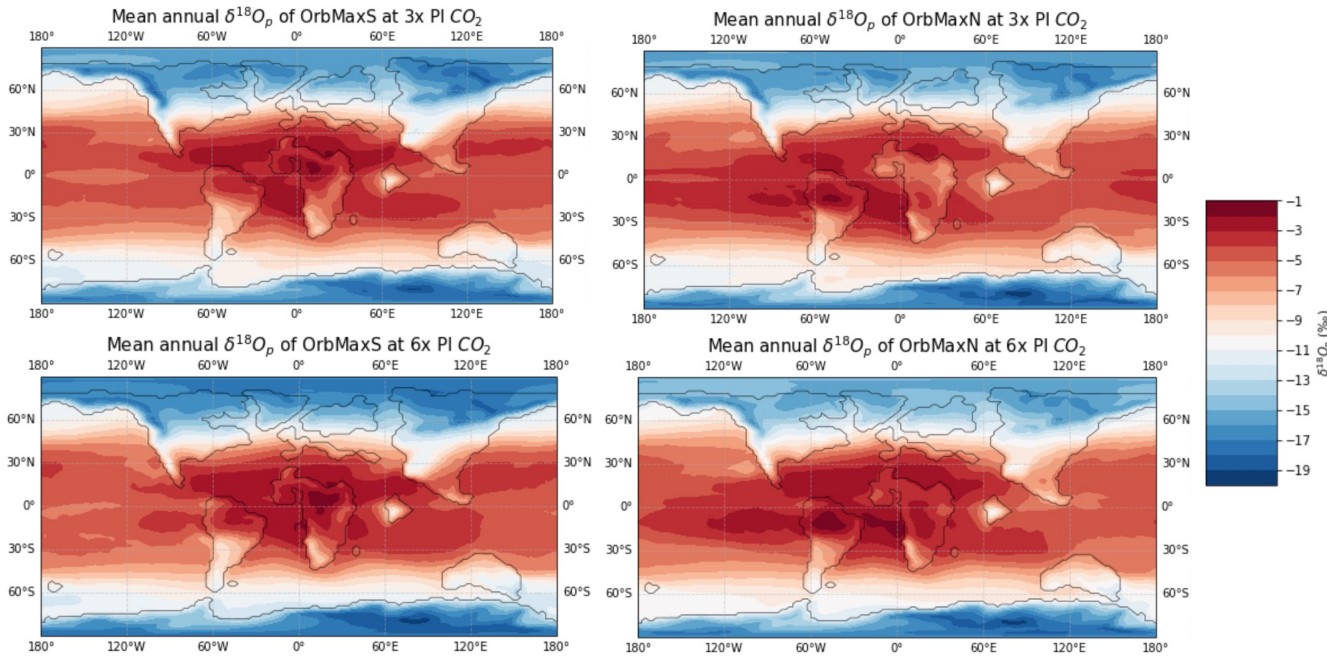

605

**Figure A13: The mean annual $\delta^{18}O_p$ at 3x PI CO$_2$ (above) and 6x PI CO$_2$ (below) for OrbMaxS (left) and OrbMaxN (right).**

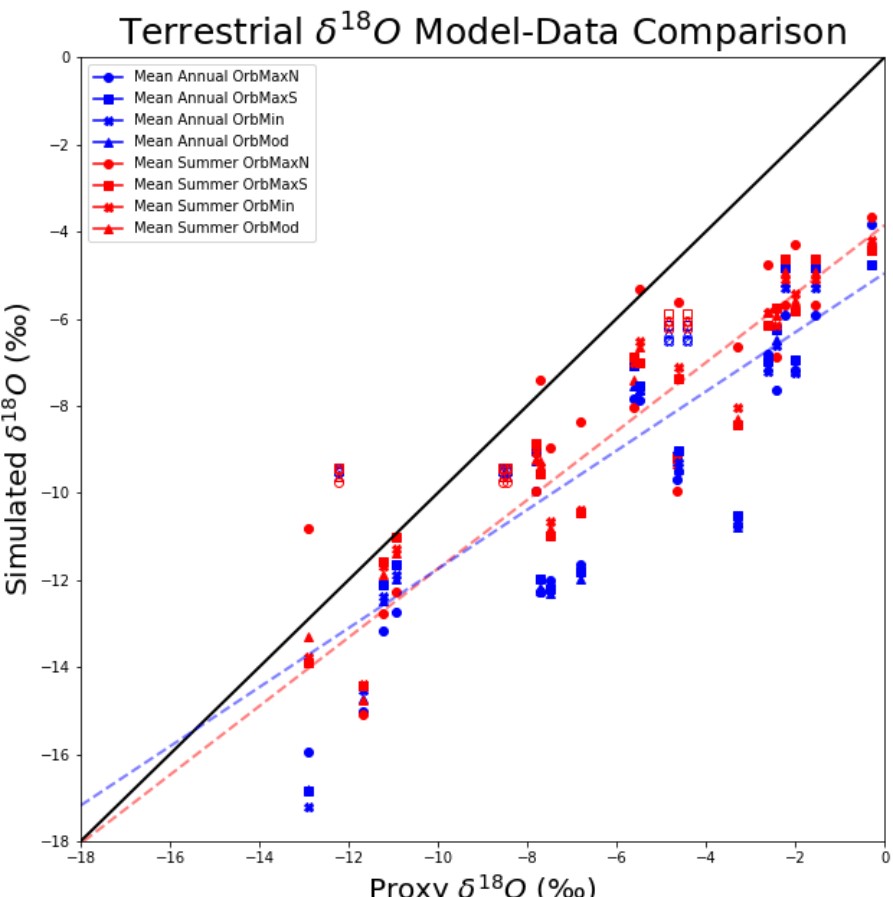

**Figure A14: The mean annual and mean summer soil water $\delta^{18}O$ at 6x PI $CO_2$ for all four orbital simulations compared to the proxy record $\delta^{18}O$. Orbit is denoted by marker shape, time of year is denoted by marker color, and unfilled markers represent paleosol carbonate records, while filled markers represent siderite records. The simulated $\delta^{18}O$ is taken from the approximate location in the model that corresponds to the location of each given proxy record. The dotted lines represent trendlines for the mean annual and mean summer values. The mean annual slope (blue) is 0.679 and the mean summer slope (red) is 0.789, suggesting that simulated mean summer isotopic signals are in closer alignment to the proxy data.**

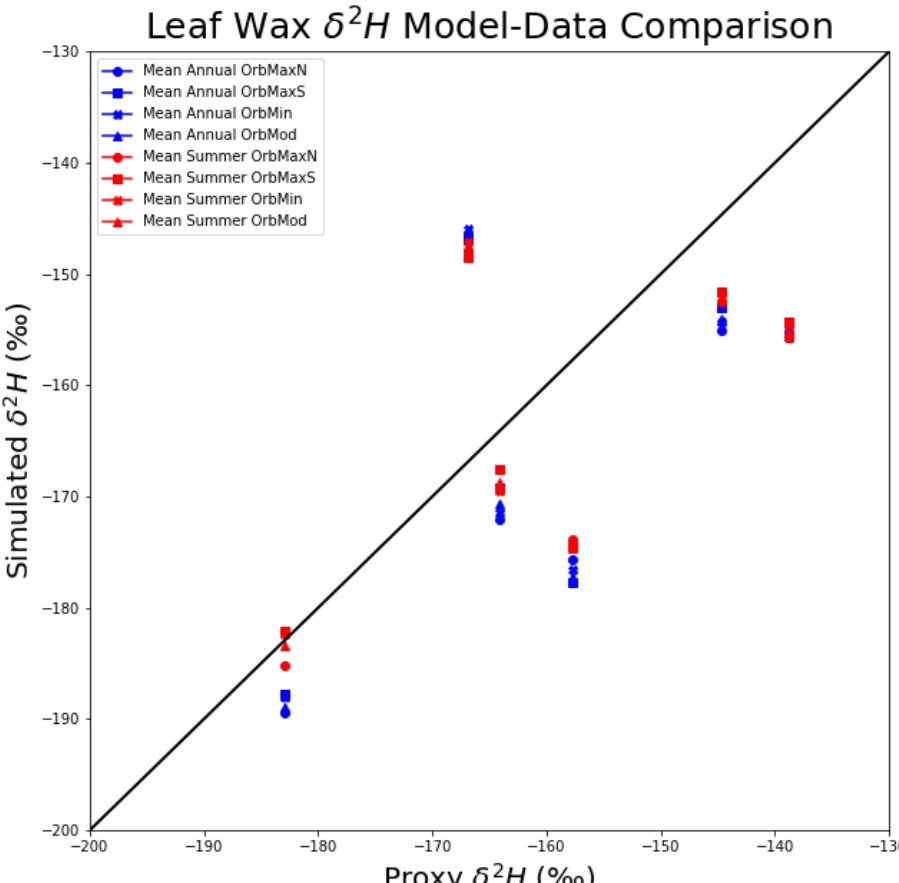

**Figure A15: The mean annual and mean summer model-inferred leaf wax $\delta^2$H at 6x PI CO$_2$ for all four orbital simulations compared to the leaf wax record $\delta^2$H. Orbit is denoted by marker shape and time of year is denoted by marker color. The simulated $\delta^2$H is taken from the approximate location in the model that corresponds to the location of each given proxy record.**

620

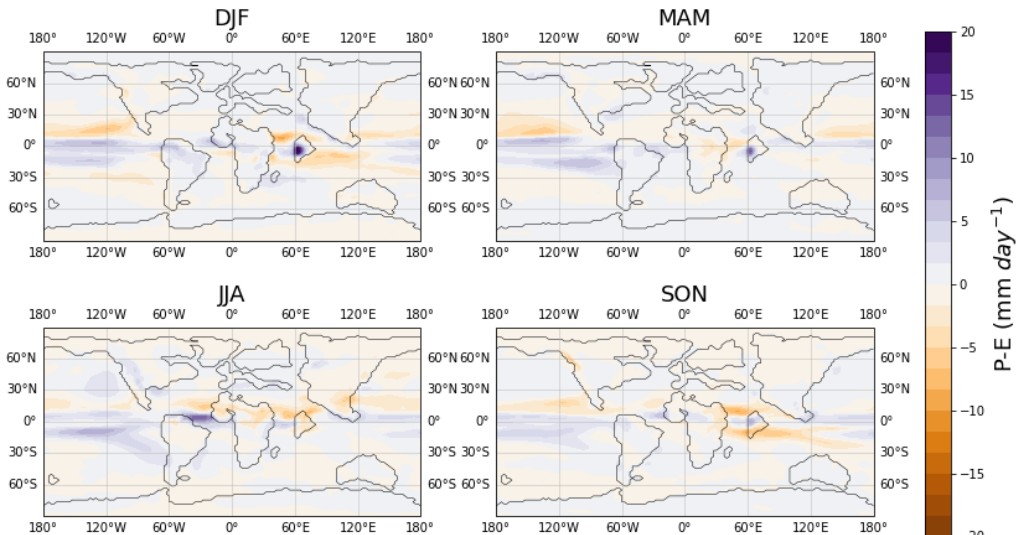

**Figure A16: The seasonal net precipitation (precipitation minus evaporation) differences between OrbMaxS and OrbMaxN at 3x PI $CO_2$.**

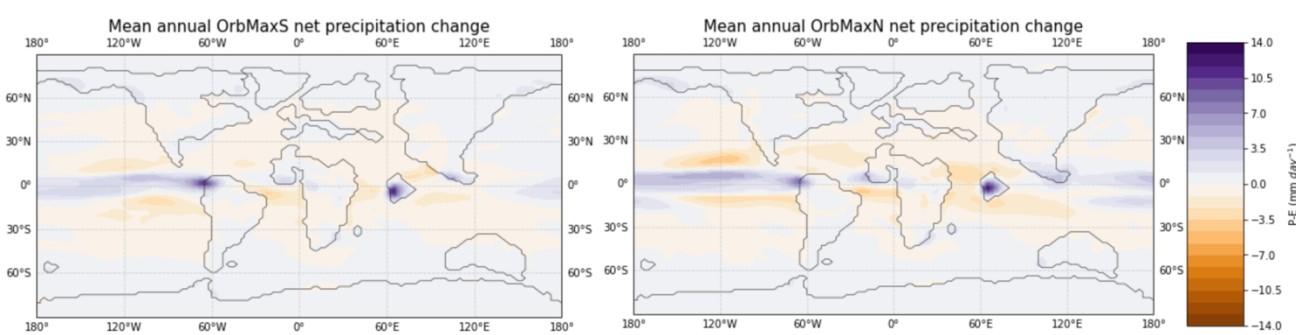

625

**Figure A17: The difference in mean annual net precipitation (precipitation minus evaporation) under 6x PI $CO_2$ compared to 3x PI $CO_2$ for OrbMaxS (left) and OrbMaxN (right).**

## Code and data availability

630

All data needed to evaluate the conclusions in the paper are presented in the paper. The published proxy records used in this study are all cited. Additional data related to this paper are available in the Zenodo repository (doi: 10.5281/zenodo.7971738). Additional data are provided on request to the authors. The CESM model code is available through the National Center for Atmospheric Research software development repository

635 (https://svn-ccsm-models.cgd.ucar.edu/cesm1/exp_tags/pcesm_cesm1_2_2_tags/dt-cesm1.0_cesm1_2_2_1/).

## Author contributions

JC and CP designed the experimental approach. JC analyzed the results and prepared the figures. JZ developed the model code and performed the simulations. JT provided proxy data. JK provided code templates for analysis. JC prepared the manuscript and all co-authors provided comments.

## Competing interests

The authors declare that they have no conflict of interest.

## Acknowledgements

This research has been supported by the National Science Foundation (grant no. 2309580). We would like to thank the editor Yannick Donnadieu, as well as the reviewers, Gordon Inglis and an anonymous reviewer, for their help in improving the manuscript.

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
