# Peer review of "CO2- and orbitally- driven oxygen isotope variability in the Early Eocene"

_Climate of the Past, 2023_

## Author Response (AR1)

**Comments made by Reviewer #1, Gordon Inglis (summarized):**

- What are the implications for orbit control on different time periods

The authors added a few sentences in the discussion section on the implications of these findings on other time periods: "Additionally, we find that orbital variability has relatively greater influence on precipitation isotopes under a lower $CO_2$ condition. This may imply that orbit exerts more control on the seasonal hydrological cycle in colder climates than warmer climates. As such, it may be especially important to incorporate the potential influence of orbital variability on colder, long-interval climate studies in future work. Studying orbital control on the hydrological cycle in warmer climates is still recommended, but it may have slightly less considerable of an impact in extremely high $CO_2$ environments."

- Additional literature that was suggested (Anagnostou, Inglis, Piedrahita, Cramwinckel, Handley, Pagani)

Several additional papers were cited throughout. We cited Anagnostou et al., 2020 and gave the range of proxy values for the Early Eocene $CO_2$ estimates. We cited Inglis et al., 2022 to potentially explain their variation in leaf hydrogen isotopes with orbital variability. We cited Piedrahita et al., 2022 for insights on orbital variability during the PETM. We cited Cramwinckel et al., 2023 as another study where Early Eocene and PETM CESM simulations are in close agreement with proxies. We cited Handley et al., 2012 and Pagani et al., 2006 for insight on leaf wax hydrogen isotope shifts with fractionation factor uncertainty.

- Use $\delta^2H$ instead of $\delta D$

Throughout the paper, $\delta D$ has been changed to $\delta^2H$.

- Possible diagenetic control on terrestrial data

Diagenesis is included as a potential source of uncertainty in the terrestrial data.

- Clarify the 13 degrees C value for PETM temperature

This was originally meant to be an absolute maximum (a small area near the poles increased in temperature by as much as 13 degrees C), but due to the confusion, this sentence was changed to be a mean annual temperature increase of 6 degrees C instead of an absolute maximum.

- Be more specific in language on changes

More specific details to the language were added throughout. For instance, "...one record is closer…" has become "...the Jaramillo et al (2010) record is closer…", "...does not experience much of an increase…" now has "~2℃ or less" added, and "...slight increase…" now has "~3% or less" added.

- Importance of fractionation factor

This is further emphasized and the maximum potential shift in $\delta^2H$ values (in consequence of uncertainty on the fraction factor) is now included: "Plus, the fractionation factor used in the WaxPSM is a globally averaged estimate, and there's a wide range of potentially realistic fractionation factors that could shift leaf wax $\delta^2H$ values by as much as ~20‰ (Handley et al., 2012; Pagani et al., 2006)."

- Inglis 2022 leaf wax record

The variability of this record is now mentioned in the discussion section: "For instance, the variation in the terrestrial $\delta^2H$ leaf record in Inglis et al (2022) may be partly attributed to orbital variability."

- Clarify dotted lines on Figures 9 and 10 and add error bars to Figure 10

The description of these figures has been expanded to further explain what the dotted lines represent, and the small uncertainty bars are now added to the figure with leaf wax records.

- Add site names to Table A1

Site names have now been added to this table.

**Comments made by Reviewer #2, Anonymous Referee (summarized):**

- Sensitivity experiments are limited in a transient context

A few sentences in the abstract and introduction have been added to clarify that these sensitivity experiments aren't meant to fully capture a transient event like the PETM, but rather are used to mimic the envelope of the event as a background time interval in order to study climatic changes as a result of manipulating $CO_2$ and orbit in a past warm climate. The data used is not dated to a specific time or orbit, so it is reasonable to compare it to a climatology snapshot of the PETM with orbital and seasonal range to further understand the environmental context of the records and potential biases. For instance, we added: "However, the terrestrial data is not dated to a specific orbit or season given uncertainties in the dating relative to orbital pacing, so we use it as

an approximate envelope of PETM water isotope values against the range of values from all simulated orbits and seasons."

- Discuss processes involved in water isotope changes more

Discussion on processes involved in water isotope changes has been expanded on to include more on atmospheric dynamics and air-sea interactions, as well as what we can confidently or not confidently express based on the coarse resolution of this global model. Specifically, more has been added on the effect of insolation on sea surface evaporation, the effect of wind currents over the Indo-Pacific warm pool, elevation, geography, continental recycling, and the Hadley circulation in the context of a higher atmospheric $CO_2$ climate. Most of the extra discussion on isotopic processes has been added to Sect. 3.1 and 3.2. For example, on western North America orbital changes we added: "Evaporated water from the cool Pacific Ocean travels in the prevailing westerly winds over continental North America. As the moisture ascends the mountainside, there is increased rainfall, further depleting the clouds of the heavier water isotopes, and leading to a dry and isotopically light descending air mass (Fig. 4)." On northern Africa orbital changes we added: "High rates of evaporation from the warm pool accompany the trade winds to transport relatively isotopically heavy moisture to the primarily warm and dry Sahara Desert region (Figs. A5, A16). This region has sparse vegetation and resulting low rates of evapotranspiration, so the water isotopes in precipitation are largely consequence of the evaporative source – the nearby seawater. The cooler, drier wind above the Indo-Pacific during SON passes over the warm sea and evokes higher evaporation rates due to the strong gradients in temperature and moisture between the air-sea surfaces, and that enriched air mass is quickly swept away towards the nearby land mass. The little rainfall this region experiences is therefore isotopically heavier in the OrbMaxS simulation (Fig. 4)." On subtropical $CO_2$ changes we added: "The narrowing tendency is largely due to the enhanced meridional moist static energy gradient seen in warming climates with increased atmospheric moisture (Fig. 6; Byrne and Schneider, 2016). Moreover, there is often a widening of the Hadley circulation projected in warming climates, which contributes to the expansion of a dry descent region off the tropics (Figs. 6, A17; Byrne and Schneider, 2016). With stronger, unsaturated downdrafts, there is an expected increase in $\delta^{18}O_p$ over most subtropical land (Fig. 7)." And on northern Africa $CO_2$ changes we added: "Additionally, the prevailing trade winds are also relatively cooler and drier over the Indo-Pacific in OrbMaxN, though the Indo-Pacific warm pool remains very warm under all orbits, resulting in a stronger temperature and moisture gradient at the air-sea interface. This gradient increases evaporation rates at the source, resulting in higher $\delta^{18}O_p$ over most of the Sahara Desert (Figs. 7, A8, A17)."

- Vegetation impact on water isotope changes

There is now mention of the precession-driven modifications in global vegetation and how those changes in biosphere-atmosphere interactions would have impacted isotopes, especially in leaf waxes. However, we do not change vegetative inputs between simulations for several reasons: the global vegetation changes between the PETM and Early Eocene are not well known, the estimated fractionation factors have high uncertainty, and the isotope-enabled land model assumes the transpired water has the same isotope ratio as the root-weighted soil water. A paragraph has been added to the introduction to explain this. Mentions of transpiration and continental recycling have been added as well, though these processes are not a focus in this paper as vegetation remained the same between simulations. There is also further clarification on what exactly is manipulated between simulations ($CO_2$ and orbit) so any changes in water isotopes can be attributed to those factors, rather than changes in global vegetation as we want to isolate the impact of $CO_2$ and the impact of orbit on the terrestrial water cycle.

- Why the Early Eocene / absence of cryosphere

The reason the simulations are of Early Eocene geography and climate are clarified in the introduction, along with the significance in a lack of cryosphere. For instance, after explaining that atmospheric $CO_2$ will exceed 1000 ppm by the end of this century under the higher emissions pathway, which the Earth hasn't experienced since the Early Eocene, we added: "This study serves to distinguish the warming signal from the orbital signal within the hydrologic cycle under the most recent extreme warmth." As for a lack of cryosphere, we added: "Warmer summers often melt more ice which can accelerate a climatic fluctuation, but the warm Early Eocene lacks a cryosphere, which may have modified the climate's response to warmer summers."

- References (Berger, Ruddiman, Craig, etc)

References mentioned as possibly inappropriate given the context have been modified or deleted, except for Craig (1961) as that reference is given in relation to a short, general description of water isotopes and Craig (1961) is the pioneer paper on this subject and still relevant to this discussion on water isotopes.

- Is the water isotope simulated in ocean model

Yes, this has been clarified: "These simulations track water isotopes in every component of the model (Brady et al., 2019)."

- Difficulties in interpreting terrestrial data

Limitations concerning the data are discussed in the methods and discussion sections, including that the data are not dated to a specific orbit. The addition of two point-by-point figures (Figs. A14, A15) comparing simulated and proxy isotopic signals at each record's location allows for stronger comparison and mention of regional effects, though local effects are not necessarily simulated by a low resolution model. More details on limitations and biases in the proxy records are included: "The summer season receives the greatest insolation, which increases temperature and evaporation rates, which in turn would have biased the isotope recording if this environment did encourage faster soil carbonate growth. This bias is seen in the point-by-point comparison as well, which highlights regional climate over global climate, since the simulated mean summer isotopic signals more closely mimic the proxy data than the simulated mean annual isotopic signals (Figs. A14, A15)." As well as potential biases in the model itself: "This slight low bias may partially explain the misalignment between simulated and proxy isotopic signals that is seen in the point-by-point comparison as well (Figs. A14, 15). The model also exhibits some depleted bias in $\delta^2 H$, as well as the presence of a double ITCZ, which may encourage biases in extratropical moisture transport (Brady et al., 2019)."

- How the simulations were run / how they have been used in the past

Information regarding vertical levels in the model, where water isotopes are tracked, the strengths and weaknesses of this model and therefore what it is best used for, isotope biases within the model, how these simulations have been used in the past, and how this paper offers new information has been clarified in the introduction and methods sections. For example, we added: "These simulations track water isotopes in every component of the model (Brady et al., 2019). This Early Eocene model has been previously published in Zhu et al (2019), Zhu et al (2020), and Tierney et al (2022). The simulations at varied $CO_2$ have been used to analyze ocean circulation and shortwave cloud feedbacks to further understand parameterizations within the model that play a role in large-scale climate sensitivity (Zhu et al., 2019; Zhu et al., 2020). Additionally, these simulations have been statistically sampled through a data assimilation approach in order to reconstruct PETM climate changes (Tierney et al., 2022). None of these previous studies investigate the exceptional variations in seasonal climate between orbits or the sensitivity of the terrestrial water cycle to both orbital and atmospheric $CO_2$ changes. Through tracking $\delta^{18} O_p$, this paper underlines the importance of orbital cycles in understanding terrestrial water isotopes."

- Methane concentration

We clarified in the methods section that methane concentration was held constant at PI levels but it may have been higher in these climates: "Atmospheric greenhouse gases other than $CO_2$, like $CH_4$, may have been higher in a warmer climate, but these are kept identical between

simulations. Greenhouse gases other than $CO_2$ are poorly constrained for the Early Eocene and the warming effect on the water cycle is largely captured by the change in atmospheric $CO_2$."

- What is the paleo calendar effect

We added a sentence generally describing how this was used: "The modeled summer days have been adjusted for the paleo-calendar effect, which structures time as a fixed number of degrees in Earth's orbit, rather than a fixed number of days each month, so that seasonal comparisons between simulation and data are properly lined up according to the Earth's position in its orbit (Bartlein and Shafer, 2019)."

- What is the microclimate point

Any mention of microclimate has been removed as it was not relevant to the simulations anyway given the low resolution.

- Seasonal bias and insolation

The effect of insolation on warm-season biased water isotope proxies has been further clarified in the discussion section.

**Comments made by Editor, Yannick Donnadieu (summarized):**

- Clarifications of processes underlying changes in oxygen isotopes in the simulations

As advised by both Reviewer #2 and the editor, more detail on the processes that govern changes in oxygen isotopes and regions where we see drastic changes have been added. Details on processes affecting water isotopes globally in Figures 4 and 7 have been added to the results section as well. For instance: "The increase in insolation during DJF also encourages stronger evaporation rates from the sea surface, which is influential on isotopic signals as the origin of transported moisture, and sometimes encourages continental recycling through evapotranspiration (Figs. 4, A16; Gierz et al., 2017; Risi et al., 2019). With higher insolation and lower relative humidity, the air is generally drier and able to stimulate higher rates of evaporation (Fig. A5). Therefore, the DJF season experiences higher insolation, warmer temperatures, higher rates of evaporation, and generally a stronger presence of heavier isotopes in atmospheric moisture and, in turn, precipitation (Figs. 2, 3, 4, A3, A5). JJA experiences a decrease in insolation, lower temperatures, lower rates of evaporation, and generally a weaker presence of heavy isotopes in precipitation (Figs. 2, 4, A1, A5)." These details mostly focus on atmospheric dynamics and the air-sea interface; smaller, localized processes that may have affected the proxy records are mentioned but not a focus due to the coarse resolution of the model.

- The way the simulations have been done

More details on the simulations have been added to the introduction and methods sections. Clarifications on the number of vertical levels of the atmosphere and ocean, where the simulations track water isotopes, what known isotope biases exist within the model, what inputs were changed between simulations and what was identical between simulations, and largely what the simulations are best used for have been included.

- Clearer take home message

We added sentences in the introduction that clarify the take home message, as well as in the discussion - regarding orbital controls under different $CO_2$ levels and the relevance to proxy data studies, and in the conclusion. For instance, in the introduction: "By modeling different orbital and $CO_2$ configurations of the Early Eocene, and matching the simulations to fossil evidence, we can provide context for the proxy records and learn how the orbit may have played a part in the severe warming at the onset of the Paleocene-Eocene Thermal Maximum (PETM), as well as different seasonal impacts on the Early Eocene climate. This study serves to distinguish the warming signal from the orbital signal within the hydrologic cycle under the most recent extreme warmth." And in the conclusion section: "These findings highlight the importance of modeling various orbital states to understand variation in water isotope records and stress the influence changes in orbit have on seasonal climate relative to changes in greenhouse gases. This study also determines that the concentration of greenhouse gases in the atmosphere partly controls the sensitivity of the climate to orbital changes."

- Clearly state how these simulations have been previously used and why this paper is different/important

A few sentences have been added to the introduction that explain what previous publications use some collection of these simulations (Zhu et al., 2019; Zhu et al., 2020; Tierney et al., 2022) and how those simulations were used in those studies. For example: "This Early Eocene model has been previously published in Zhu et al (2019), Zhu et al (2020), and Tierney et al (2022). The simulations at varied $CO_2$ have been used to analyze ocean circulation and shortwave cloud feedbacks to further understand parameterizations within the model that play a role in large-scale climate sensitivity (Zhu et al., 2019; Zhu et al., 2020). Additionally, these simulations have been statistically sampled through a data assimilation approach in order to reconstruct PETM climate changes (Tierney et al., 2022). None of these previous studies investigate the exceptional variations in seasonal climate between orbits or the sensitivity of the terrestrial water cycle to both orbital and atmospheric $CO_2$ changes. Through tracking $\delta^{18}O_p$, this paper underlines the importance of orbital cycles in understanding terrestrial water isotopes."

- Point-by-point comparison rather than a zonal mean as shown in Figure 9

The original point-by-point comparison was done as an RMSD. The latitudinal figure was able to portray whether the simulations captured the data or not, the expansive seasonal range of the highest monthly mean and lowest monthly mean of oxygen isotope signals, whether the data appears seasonally biased or not given this seasonal range, and how the simulated isotopic signals vary latitudinally. However, visualizing the point-by-point comparison can confirm the seasonal bias suspicion in the records by taking regional effects into account, rather than just global. Therefore, I added an alternative to Figures 9 and 10 in the appendix (Figs. A14, A15) where the simulated mean annual and mean summer isotopic values are taken from the approximate location of each proxy value. Those values are plotted with the proxy record values alongside a 1:1 line - the closer the values are to the 1:1 line, the closer the simulated values are to the empirical data. These figures are still able to portray the similarity in the isotopic records to the simulated values, include all four orbits, and include both a mean annual and mean summer value for every location and simulation to convey possible seasonal bias. I was unable to make these figures more simple and concise without removing information that I wanted included. They support the conclusions from the original figures but are a bit more tortuous, so I have added them to the appendix rather than replaced the main figures.

- Check out the paper published by Tardif et al. (2021, Sci Adv)

This citation was added where it was relevant, following a sentence on precession-driven vegetation changes impacting the hydrologic cycle isotopes: "The Earth experienced a major precession-driven modification of global vegetation during the PETM and across the Eocene, so the changes in biosphere-atmosphere interactions and plant biology could have significantly impacted the hydrological cycle and leaf wax isotopes (Tardif et al., 2021)."